



# Biomass burning events measured by lidars in EARLINET. Part I. Data analysis methodology.

Mariana Adam[1], Doina Nicolae[1], Iwona S. Stachlewska[2], Alexandros Papayannis[3], Dimitris Balis[4]

[1]National Institute for R&D in Optoelectronics, Magurele, 077225, Romania
[2]Institute of Geophysics, University of Warsaw, 02093, Poland
[3] National Technical University of Athens, Department of Physics, Athens, 15780, Greece
[4] Laboratory of Atmospheric Physics, Aristotle University of Thessaloniki, Thessaloniki, 54124, Greece

*Correspondence to*: Mariana Adam (mariana.adam@inoe.ro)

**Abstract.** The methodology of analysing the biomass burning events recorded in the database of the European Aerosol
Research Lidar Network in the frame of Aerosol Cloud and Trace Gases Research Infrastructure is presented. The period of
2008-2017 was chosen to analyse all of the events stored in the database under Forest Fire category for a number of 14 stations
available. The data provided ranged from complete data sets (particle backscatter, extinction and linear depolarization ratio
profiles) to single profiles (particle backscatter coefficient profile). Smoke layers geometry was evaluated and the mean optical
properties within each layer were computed. The backtrajectories technique was used to double check the source of all pollution
layers. The biomass burning layers were identified taking into account the presence of the fires along the backtrajectory. The
biomass burning events are analysed by the means of the intensive parameters. The analysis was structured in three directions:
I) common biomass burning source (fire) recorded by at least two stations, II) long-range transport from N. America, III)
analysis over four geographical regions (SE Europe, NE Europe, Central Europe and SW Europe). Based on backtrajectories
calculations and fires' location, the lidar measurements can be labelled either as measurements of 'single fire' or 'mixed fires'
(case I), measurements of N America fires or measurements of mixed N America and local fires (case II). The histogram of
the fires' location reveals the smoke sources for each region. For each region, statistics on intensive parameters is performed.
The sources' origin of the intensive parameters is categorized based on the continental origin of the air-mass (European,
African, Asian, N American or a combination of them). The methodology presented here is meant to provide a perspective to
explore large amount of lidar data and deliver novel approaches to analyse the intensive parameters based on the assigned
biomass burning sources. A thorough consideration of all potential fires' sources reveals that most of the time the lidar
measurements characterise the smoke from a mixture of fires. A comprehensive discussion over all results (based on the
intensive parameters and the sources' location), will be given in a companion paper submitted to ACP EARLINET special
issue.



## 1 Introduction

Biomass burning (BB) represents one of the major sources of atmospheric particles (aerosols). The sources of BB are both natural (wild fires) and anthropogenic (controlled fires). The direct effect on radiative transfer can be both negative and positive as a consequence of the opposite effects of the two main components: black carbon (absorption, positive effect) and organic

carbon (scattering, negative effect) (Fiebig et al, 2002; Bond et al., 2013; Myhre et al., 2013) with a general radiative forcing close to zero (https://www.ipcc.ch/site/assets/uploads/2018/02/WG1AR5_Chapter08_FINAL.pdf, IPCC, 2013, last access 20200203). As indirect effect, BB can act as cloud condensation nuclei (e.g. Yu, 2000) or ice nuclei (e.g. Prenni et al, 2012). At ground level or within the Planetary Boundary Layer (PBL), strong pollution events produce a large reduction of the visibility over various regions (e.g. Adam et al., 2004), which can affect the traffic and more importantly can cause serious

health issues for humans (e.g. Pahlow et al., 2005; Sapkota et al., 2005, Alonso-Blanco et al., 2018). Numerous wild forest fires events occur each year in Europe (refer to annual reports from European Commission on "Forest Fires in Europe, Middle East and North Africa YYYY" at http://effis.jrc.ec.europa.eu/reports-and-publications/annual-fire-reports/ available since 2000, where YYYY represents the year; last access 20191126). Therein, the studies report the impact of forest fires in Europe, providing the number of fires and burnt area by country. The 2017 report (San-Miguel-Ayanz et al, 2018) shows that the

number of fires was higher as comparted with the average over 2000-2017 period while the burnt area was even larger, especially for Portugal. For the Southern States (Portugal, Spain, France, Italy and Greece), most effected by fires, the highest number of fires occurred in Portugal (44%) and Spain (29%) while the largest burn area occurred in Portugal (59%) and then Spain (19%) and Italy (18%). As mentioned in the report, climate change affects forest fires through the weather conditions, vegetation and fuel. Besides the European fires which can act locally or through short range transport within continent, Europe

is also affected by long range transport, the most common being the smoke transported from Russia or even from North America (aged smoke). Note that over Poland the BB is the most important source of long range transported particles even in boundary layer (e.g. Wang et al., 2019). In a recent study by Nicolae et al. (2019) it was shown that the smoke is the most predominant aerosol type in Europe for long-range transport, as revealed by lidar measurements over 2008-2018 period over 17 stations equipped with multiwavelength lidars. The inventory of fires is provided by European Commission through EFFIS

(European Forest Fire Information System) at European level and by NASA through FIRMS (Fire Information for Resource Management System) at global level (https://firms.modaps.eosdis.nasa.gov/, last access 20191126) (Davies et al., 2009). Both databases are based on MODIS (Moderate Resolution Imaging Spectroradiometer) satelitare data.

Lidars provide measurements of the smoke aerosol being able to deliver the boundaries of the smoke layers as well as the optical properties of the smoke aerosol in the layers. Both layers geometry and optical properties can be used as a validation

of the transport models (e.g. van Drooge et al., 2016). For a detailed analysis of the data requirements by models, see Benedetti et al., 2018 and references herein. Referring to BB aerosol, the study mentions the usefulness of lidar retrieval of the plume height. Layer geometry can also be used to constrain specific satellite profiles retrievals which, as known, have less sensitivity towards the ground (e.g. Bond et al., 2013). The optical properties can be used (closure studies) to model the microphysical



properties of the aerosol (e.g. Fiebig et al., 2002; Fiebig et al., 2003). Regarding the impact of BB aerosol on weather forecast, the study by Zhang et al. (2016) suggests that the BB' effect is seen for AOD (aerosol optical depth) at 550nm larger than 1. Ultimately the researchers try to establish the BB effect on the radiative forcing (e.g. Fiebig et al., 2002; Lolli et al., 2019; Markowicz et al., 2016).

Within these applications, the EARLINET (European Aerosol Research Lidar Network) can provide spatial and temporal coverage of the BB transport over Europe. EARLINET is part of Aerosol Cloud and Trace Gases Research Infrastructure (ACTRIS) (http://actris.nilu.no/, last access 20191126), which provides ground-based, insitu or remote sensing. EARLINET provides measurements of the biomass burning aerosol with first lidar profiles dating from 2002. More systematic measurements over the network can be seen after 2012 (closely related with the increase of the number of stations in the

network). A review of the lidar studies over BB will be given at a later stage in this paper.

The current study envisages the research on the biomass burning aerosol as measured by 14 stations in EARLINET over the period 2008-2017. This paper presents the methodology of data analysis, whereas a companion paper (submitted to ACP, EARLINET special issue) focuses in detail on the discussion of the outcome. The EARLINET is shortly presented in Section 2. In Section 3 we briefly discuss the extra undertaken data quality checks. Data analyses follows in Section 4. The data

interpretation (Section 5) is based on aerosol intensive parameters (lidar ratio - LR, extinction Ångström exponent - EAE, backscatter Ångström exponent - BAE, linear particle depolarization ratio - PDR). Examples are given for the following research directions: I) fires events (same source) as observed by two lidar systems, II) long range transport (LRT) from N America, III) study based on four geographical measurements regions including the histogram of the fire sources locations for each region. Summary and conclusions follow in Section 6. A short introduction to the present methodology was given at

ILRC29 (Adam et al., 2019).

## 2 EARLINET database

The EARLINET database consists currently (October 2019) of 30 lidar stations covering most of Europe and one location outside Europe (Dushanbe, Tajikistan). There are three other non-permanent stations as well as six newly joining stations. Twelve stations are not active (https://www.earlinet.org/index.php?id=105, last access 20191126). A review of the EARLINET

network is given by Pappalardo et al. (2014). The data submitted to the database belongs to one or more of the following categories: climatology (measurements taken at specific times on Mondays and Thursdays), Calipso (measurements taken during Calipso overpasses), Saharan dust (measurements taken during Saharan dust intrusions), volcanic eruptions (measurements taken during volcanic eruptions), forest fires (measurements taken during forest fires events), diurnal cycles (measurements taken continuously over the whole day), Cirrus (measurements of Cirrus clouds), photosmog (measurements

during photosmog episodes), rural/urban (measurements taken under rural or urban environments) and stratosphere (measurements at stratospheric levels). Most of the measurements are taken under Climatology category followed by Calipso and Saharan dust categories (Pappalardo et al. 2014). The largest data set regarding the optical properties was submitted by





each station following its own retrieval procedure. Note that while Climatology and Calipso measurements are mandatory in the network, all the other categories are built based on voluntary bases. Currently it is envisaged the use of Single Calculus Chain (SCC) developed in EARLINET for all raw data submitted by each station (see D'Amico et al, 2015; D'Amico et al, 2016; Mattis et al, 2016). There are two types of files submitted to the database for the optical properties: b-files (containing

particles backscatter coefficient) and e-files (containing particles extinction coefficient and sometimes particles backscatter coefficient retrieved using both elastic and Raman channel). Other variables may be reported as well (see Pappalardo et al., 2014). For systems with depolarization capability, the retrieval of the particles linear depolarization ratio (PDR) is recorded in b-files except Warsaw ("wa"), which records it in both b-files and e-files.

## 2.1 EARLINET Forest Fire category

For the present study, the data submitted under Forest Fire (FF) category was considered. Over the period 2008-2017 (when most of the data was submitted), a number of 3759 b-files and e-files were available at 20 stations corresponding to the emission wavelengths 355, 532 and 1064 nm. Another 256 files were submitted at different emission wavelengths (313, 351, 510 and 694 nm). In addition, the Hysplit (Hybrid Single-Particle Lagrangian Integrated Trajectory model) model was available since September 2007 (Stein et al., 2015; Rolph et al., 2017). Hysplit model is used to double check the origin of the aerosols from

various layers (as discussed later on). We also considered only the data whose emission wavelength was at 355, 532 and 1064 nm as these are the most used in the lidar community and allows the direct intercomparing of the optical properties. After preliminary quality control checks (discussed in chapter 3), 14 stations were selected over 2008-2017 period, delivering 2341 files (~ 60 % of total), quality checked. The geographical location of the stations is shown in Fig. 1. Also shown, the four geographical regions analysed separately further on (section 5.3). An overview over the number of stations reporting data for

FF category as compared with their total number of files submitted (as for March 2018), reveal that the following. FF category represented ~ 40% of the total measurements for Thessaloniki, Athens and Warsaw, ~ 20 % for Minsk and ~ 10% for Bucharest and Granada.

Several stations ("at", "ba", "bu", "le", "po", "oh", "th", wa") were committed to send either reprocessed data or to process additional data using SCC, which were not yet included yet in EARLINET (most of them for 2017) as for March 2018. Within

these circumstances, the number of files increased to 3589 (QC for 2341 files already on EARLINET database). The diagram of the methodology is shown in Fig. 2. Following I$^{st}$ step in methodology (Fig. 2), the number of input files (b-files and e-files) for each station is shown in Fig. 3 and Table 1. As a first remark, we can observe that we handle a large variety of lidar systems. Thus, we have one system with 3b+2e+2d ("wa"), three systems with 3b+2e+1d ("bu", "oh" and "po"), five systems with 3b+2e ("at", "ca", "ev", "gr", and "th"), four systems with 3b ("ba", "be", "le" and "mi") and one system with 1b ("sf"). The

total number of b-files and e-files is shown in Table 1. The total number of particles backscatter coefficient, extinction coefficient and linear depolarization ratio after the first quality checks (EARLINET checks) is: #βp355=969, #βp532=864, #βp1064=830, #αp355=524, #αp532=503, #δp355=131 and #δp532=320 (total of 4141 profiles). The distribution by stations





follows closely Fig. 3. The stations with the largest contribution for Forest Fire category are Athens, Bucharest, Granada, Thessaloniki and Warsaw.

Note that the number of particle extinction coefficient profiles equals the number of "e" files while the number of particles backscatter coefficient at 1064nm equals the number of b1064 files. In the case of particles backscatter coefficient at 355nm

and 532nm, their number can be larger than the number of "b" files as retrievals of backscatter coefficient is reported also in "e" files, employing the use of Raman channel in the retrieval of backscatter coefficient (using either Raman method or aerosol backscattering ratio method). As seen in Table 1, this is the case for the stations "bu", "gr", "po" and "wa". For the same time stamp, the profile from "e" file is kept. The reason behind is that the backscatter coefficient will be further used to calculate the lidar ratio, and the same spatial resolution as the extinction coefficient is desirable. The occurrence in time of the retrievals

is shown in Fig. 4. Only the cases for which the layers have a fire origin are shown (smoke layers). As observed, there are sporadic measurements reported for the years 2008-2011. Starting with 2012, more stations started reporting biomass burning pollution events. The total number of time stamps for which we have at least one profile with an optical property is shown in the last column of Table 1.

## 2.2 Overview of the metadata

Metadata refers to those parameters which give information about the instrument and site location, data processing or analysing, and are saved as general attributes in the "e" and "b" netCDF files. The fixed parameters refer to the measurement position, system name, emission and detection wavelengths. The dynamic parameters change in time and space: date and time of measurement, raw resolution, time resolution (shots averaged), detection mode, final resolution, evaluation method. Another two parameters (Input Parameters and Comments) can give additional information (e.g. values used for Rayleigh calibration).

Detection mode (type of data acquisition) can be analog, photon counting or both (glued signal). Evaluation method (approach used to retrieve the optical properties) can be Fernald-Klett method (used in b-files to calculate particle backscatter coefficient), Raman method (used in e-files to determine particles extinction coefficient), aerosol backscatter ratio to obtain particles backscatter coefficient using a combination of both elastic and Raman channels (recorded in e-files). The information for parameters is provided in a free style comment format and we could not perform the quantification for all parameters (e.g. we

could not determine which was the final spatial resolution nor the method used for smoothing and error calculation). In general, there are three approaches reported: 1) keep the errors within an established threshold and thus vary the spatial resolution, 2) maintain a fine spatial resolution (with high statistical errors), 3) perform variable smoothing (e.g. lower spatial resolution above PBL). As expected, the backscatter profiles are less smooth than the extinction profiles.

The summary of the statistics (based on entire dataset input) over the Detection mode, Evaluation method, Raw resolution,

Shots averaged and Zenith angle for each backscatter and extinction coefficient is given in Table 2 (Fig. 2, stage III in Methodology). The main features are the following. For b355 and b532 files, the detection mode the most employed is the glued signal, while for b1064 it is the analog signal. For e-files, the detection mode the most employed is the photon counting. The most used evaluation approach for b-files is Klett-Fernald (for particles backscatter coefficient) while for e-files is Raman



(for particles extinction coefficient). Please note that there are 139 cases reported for b1064 which uses as evaluation method Raman ("wa" station). The station used a PollyXT system and the algorithm employed to calculate the particles backscatter coefficient at 1064 nm involves the extinction coefficient determined at 607 nm from the Raman signal (no reference is available). Concerning the raw resolution, most of the stations used 3.75 m for backscatter and 60 m for extinction. The zenith

angle was mostly 0. The shots averaged were in general of tens thousands. This overview shows that we deal with a variety of approaches to compute the mean profile and associated uncertainty.

**3 Data quality control**

The first check on data quality consisted of verifying the data which passed the quality control a.t. EARLINET criteria (Fig. 2, stage II in Methodology). The description of the checks can be found on EARLINET website

(https://www.earlinet.org/index.php?id=125, last access 20191126). Basically, there are two types of checks. First check consists on technical checks on the raw data submitted (applied on-fly since June 2018). Here the conformity of the file content is checked w.r.t. EARLINET file structure and quality control (QC) procedures. With on fly version, an automatic feedback is provided to the Data Originator reporting all the problems incurred for each rejected file, fostering the prompt resubmission of the data (see 20180614 report on https://www.earlinet.org/index.php?id=293, last access 20191126). Also, the files not

providing the associated error are eliminated. The second check (physical), applied off-line, first verifies if the data was submitted to the right category. Then, the errors are checked for negative values occurrence. Specific checks on backscatter and extinction coefficients are performed (e.g. if larger than a threshold, if integrated quantities such as AOD are within common limits) as well as on intensive variables (where possible), such as LR. The current short description corresponds to the documents issued by 14 June 2018. Three text files are generated with the name of the files rejected by technical criteria

(QC_0.0.txt) or passing technical criteria but failing physical criteria (QC_1.0), the files passing both tests being shown in the file QC_2.0. After the EARLINET QC tests, there were 2341 out of 3579 initial files (available from 14 stations) that there were passing both tests from (i.e. ~ 60 %). After adding reprocessed data/newly processed data, the total number of input files increased to 3589.

Several in house checks (most of them manually performed) were further applied for the selection of good quality data. Recall

that the additional and revised profiles sent by several stations were not on-fly QC by the EARLINET procedures. Note that the additional quality checks are performed along various steps during data analysis and are briefly discussed here. The off-line QC was manually performed for each profile before applying the algorithm for the layers' detection (section 4.1). Thus, profiles showing atypical behaviour (biased towards negative values, unrealistically increasing above a certain altitude, with PDR outside the [0 1] range) were dismissed. Sometimes, there were profiles which did not show a clearly defined layer and

thus they were eliminated (section 4.1.). The profiles for which no fire was found along the backtrajectory were also eliminated (section 4.3). The optical properties in a specific layer were dismissed if the number of available points were less than 90%, and thus, no mean value was calculated (section 4.4). Values of the mean optical properties for which SNR was less than 2





were also dismissed (section 4.4). Finally, the values of the intensive parameters with SNR < 2 and outside the imposed limits are also excluded (section 4.4).

## 4 Data analysis

### 4.1 Calculation of the aerosol layers boundaries

Prior to evaluation of the layers' boundaries (Fig.2, stage IV in Methodology), a manual QC of the profiles of the optical properties was performed. Thus, a number of optical properties profiles which did not show expected realistic behaviour were eliminated. The most common behaviours of the profiles being eliminated are:

- Profiles showing a large range of negative values due to a systematic error resulting in a bias towards negative values.
- Profiles displaying an increase above a certain altitude, most probably due to a non-accurate calibration region.
- Depolarization profiles displaying values outside [0, 1] region.
- To noisy profiles (SNR < 1).

The following approach was considered to calculate the boundaries of the aerosol layers. The order of selecting the optical profile to determine the boundaries of the aerosol layers is the following: $\beta p1064$, $\beta p532$, $\beta p355$, $\kappa p532$, $\kappa p355$. In other words, when available, use $\beta p1064$. When $\beta p1064$ is not available, use $\beta p532$. If the latter is not available either, use $\beta p355$
and so on. Note that for the times when none of the profiles showed a pollution layer, all profiles for that specific time were excluded.

Once the optical profile (including the associated error profile) and the corresponding altitude profile are available, the algorithm developed to determine the aerosol layers boundaries is run. The steps of the algorithm are the following:

- Perform a smoothing of the optical profile. The number of bins used for smoothing depends on the input resolution.
Thus, for a resolution of 3.75 m, we applied moving average over 23 bins. For a resolution of 7.5 m, 15 m, 30 m, 60 m, we used 11 bins, 9 bins, 7 bins and 3 bins, respectively. For the particular cases of "ca" and "oh" systems, we applied a number of bins of 15 and 19, respectively (as the signals were very noisy). The corresponding errors were propagated.
- Employ the function *findpeaks* from Matlab ([www.mathworks.com](www.mathworks.com), last access 20191126) to find the maxima, with
the following options: the minimum distance between peaks is 300 m (*MinPeakDistance*) and the minimum peak height (*MinPeakHeight*) is as follows: 1e-7 for $\beta p1064$, 1.5e-7 for $\beta p532$, 3e-7 for $\beta p355$, 1e-6 for $\kappa p532$ and 3e-6 for $\kappa p355$. The value of the minimum distance between peaks was chosen as in Nicolae et al. (2018). If no peaks are found, the routine returns the message no layers with maximum above *MinPeakHeight*.
- Employ the function *findpeaks* to find the minima, with the following option: the minimum distance between peaks
is 300 m (*MinPeakDistance*)





- eliminate adjacent maxima if the "prominence width" (https://www.mathworks.com/help/signal/ug/prominence.html, last access 20191126) overpasses the position of the adjacent maxima
- eliminate small maxima / minima peaks which are smaller than 10% of the maximum / minimum peak
- a maximum peak should be bordered by two minima; when the first or the last minimum is missing, a criterion is used to add the missing minimum; thus, the minimum is chosen at a location (> 300 m from the first or the last maximum peak) where the optical property has the minimum value
- the boundaries of a layer are determined by two minima with the condition that there is a maximum between them

Following the criteria discussed, there can be cases when it is not possible to find any aerosol layer. Consequently, those profiles were dismissed. Additionally, a manual check was performed and for the cases with non-accurate estimation of the boundaries, the boundaries were manually corrected (~ 40 % of the cases) and sometimes, we added layers which had a maximum below the threshold of the minimum peak height. Thus, we cope with a semi-automatic algorithm. Table 3 shows the number of time stamps when it was possible to determine a layer and at least one optical property could be calculated (column 3). Recall that many profiles were dismissed manually through quality check before we apply the algorithm for layer boundary evaluation and this explain most of the "missed" cases (difference between second and third columns). The initial total number of layers, with at least one optical property (column 4) is greater than the time series (column 3) as most of the times we have more than one layer within a profile. The other columns are discussed in the next section. Overall, we were able to determine 1901 layers for 960 time stamps (out of 1138 in total).

Various authors use different criteria to estimate the layer boundaries. In most of the papers examined, the authors do not describe how they determined the boundaries of the layers. However, the boundaries can be easily identified visually (a common practice when investigating one or few cases). In a few studies it is mentioned the gradient method (Giannakaki et al., 2015; Mattis et al., 2008; Ortiz-Amescua et al., 2017; Preißler et al., 2013). When intensive parameters are available (e.g. EAE or LR), one can determine the boundaries based on intensive parameters being nearly constant in the layer (e.g. Samaras et al., 2015) or based on the ratio of elastic to Raman profiles (Vaughan et al., 2018). In situations when a few layers are visible, one can choose them as a single large layer (e.g. Ansmann et al., 2009). Our approach provides the layers boundaries in line with those shown by Ansmann et al. (2009), Janicka et al. (2017), Hu et al. (2018), Veselovskii et al. (2018).

A few examples of boundaries estimation of the (smoke) pollution layers are shown in Fig. 5. All the optical properties profiles are shown (left - particle backscatter coefficients, middle - particle extinction coefficients, left - particle linear depolarization ratio), in order to get a glimpse of how the layers look like for all profiles. First plots (a-e) show examples of automatic selection of the layers based on developed algorithm. First four are based on b1064 signal, while the fifth is based on b355 signal. The next plots (f-i) show examples where one or more boundaries are manually modified. The last plot is based on b532 signal. In example f) a layer automatically selected around 6500 m was dismissed (considered as not substantial) while in example g) the uppermost layer was manually added. The layers are shown by the grey areas.

Most of the layers detected are situated between 1000 and 5000 m altitude (typically above PBL). However, the minimum layer bottom was found at 257.5 m while the highest layer top was found at 19,8 km. Minimum, maximum and the mean layer





thickness were 300, 6862.5 and 1337.5 m. Please note that not all the layers shown here have BB origin (as this check is not performed yet).

### 4.2 Calculation of the back-trajectories

Even if the lidar data were stored within Forest Fires category in EARLINET/ACTRIS database, in many cases there are
situations when several aerosol layers are present and not all of them have a BB origin. Thus, we would like to differentiate these cases and eliminate the non-smoke layers from analysis.

Matlab and Python routines were used to automatically obtain the back-trajectories using Hysplit. The settings for Hysplit backward run are the following: input altitude (middle of the layer) is provided about sea level (a.s.l.), run time is 240 h, meteorological model is GDAS at 0.5° resolution, vertical motion is chosen as Model vertical velocity (Fig.2, stage V in
Methodology). The terrain height is saved in the output txt file. For 23 cases (during August 2010, 2012 and 2013), the GDAS meteorology was not available at 0.5° resolution and thus, a manual run was performed using a resolution of 1°. For the situations when more than three layers were available for a profile, Hysplit was run more than one time. There were 75 such cases. We performed 1036 Hysplit runs for 1901 layers corresponding to 960 time stamps.

Hysplit backtrajectory is the most used tool to track the airmass origin. For the literature review on BB, 30 papers made use of
Hysplit (references 5, 6, 10-12, 14-21, 23, 24, 28-44, 46 on S2 Table), whereas five studies use Flexpart (refs. 2-4, 24, 26 on S2 Table). The running time used in Hysplit simulations reported in literature varies from two to ten days.

### 4.3 Layers identification based on back-trajectories and fires emissions

The output from the Hysplit text files and the information about fires taken from FIRM MODIS (https://firms.modaps.eosdis.nasa.gov/, last access 20191126) were used to produce the combined plots containing information
about both the fires' presence and the back-trajectory (Fig. 2, stage VI in Methodology). The fires are represented on the plots as a function of their fire radiative power (FRP), using different colours and sizes (see legend). The location of the trajectory, each 24 h backwards, is shown by the number of hours. The lower plot shows the altitude a.s.l. of each back-trajectory versus time. The fires are chosen within 100 km and within ±1 h around trajectory points (backtrajectory data are available at 1 h temporal resolution). In other words, we assume that those fires are more likely to contribute to the transported smoke, recorded
by the lidars. When no fire is recorded along a trajectory, we catalogue that layer as of non-biomass burning origin (dismissed from further analysis). As a consequence of this criterion, a number of 283 time stamps were considered as having layers with non-BB origin. For many cases, for the same time stamp, there were equally layers of BB origin and non-BB origin. We obtained a number of 678 Hysplit-FIRMS plots (one for each time stamp). Note that the Hysplit-FIRMS plot (corresponding to a time stamp) contains all available layers with fires detected along trajectory. Examples of such plots are given at a later
stage, when discussing specific events (Figs. 12 and 14). Based on these criteria, the number of time stamps and corresponding number of layers with BB origin are shown in columns 5 and 6 in Table 3 (compare with columns 3 and 4).





It is worth mentioning that the Hysplit model does not provide the uncertainty. In order to get a possible uncertainty of an individual trajectory, a trajectory ensemble is suggested (Rolph et al., 2017). We may assume that high uncertainties in the airmass location may occur particularly over long periods of time (e.g. ten days), which in conjunction with fires location may mean a missed fire or a fire detection that was not contributing to the measurement. Drexler

(https://www.arl.noaa.gov/hysplit/hysplit-frequently-asked-questions-faqs/faq-hg11/, last access 20191126) mentions that the uncertainty is between 15% and 30%. On the other hand, FIRMS may miss some fires (especially during cloudy atmosphere). According to Giglio et al. (2016), the collection 6 MODIS has a smaller commission error (false alarm) as compared with Collection 5 (1.2% versus 2.4% respectively). The probability of fire detection (regionally) increased by 3% in boreal N America, while staying almost the same in regions as Europe or N Africa. We have been using fires with a confidence level

larger than 70%. We did not investigate the injection height based on FRP in order to estimate if the smoke of a particular fire reached indeed the altitude of the backtrajectory. We would like to emphasize that due to the satellites polar orbit, the same geographical location can be seen four times a day at the equator and more times as the latitude increases (due to orbits overlap). Thus, we may miss a certain number of fires (which burn less than few hours, between the two orbits). However, we may consider those short life fires as not significant in smoke production.

FIRMS database was used in several studies to identify the BB origin. However, all fires occurring over certain periods (for which the backtrajectories were calculated) are typically accounted for. Thus, there were reported fires occurring over the whole day (e.g. Nicolae et al., 2013; Stachlewska et al., 2018; Janicka et al., 2017), or several days (e.g. Mylonaki et al., 2017; Heese and Wiegner, 2007; Tesche et al., 2011). By contrast, our novel approach accounts only for those fires which were occurring around backtrajectory (100 km radius) at the time of air masses passage (± 1 h).

**4.4 Calculation of the mean optical properties and intensive parameters inside the layers**

After determining the aerosol layers and their boundaries (sections 4.1 – 4.3), the mean value of the optical properties is computed for each layer (Fig. 2, stage VII in Methodology). Note that the first and last 50 m of each layer were not considered for the calculation (cf. Nicolae et al., 2018). The uncertainty for each mean value was computed following the error propagation. The mean value in the layer was calculated if there were at least 90% of the points available (the ratio between

the layer depth and the resolution in the layer). There were many cases for which the extinction coefficient or the linear particle depolarization ratio could not be calculated as typically their profiles have a shorter extend than that of a backscatter coefficient. A visual check is performed as well, and thus, where the profiles were suspicious, the mean values in the layers were manually set to NaN.

The signal to noise ratio (SNR) was computed as the ratio of the mean value to its uncertainty (e.g. Nicolae et al., 2018). The

values of each optical property for which SNR < 2 were dismissed. In Table 3 (columns 7 and 8), we show the number of the time stamps and the layers with a good SNR for optical properties. As a result of this criterium, only one time stamp was dismissed ("at") and three layers (one for each of "at", "bu" and "oh"). Please keep in mind that the number of dismissed





optical properties in layers is larger than the number of dismissed layers, as one layer is dismissed only when all optical properties in the layers are discharged.

Figure 6 shows an example (for "at") of the number of layers selected and the corresponding number of optical properties evaluated in each layer. For this example, we have 171 / 172 time-stamps from which we could determine 250 layers. It was

just one layer for which we could not determine the optical properties due to SNR constraint (see columns 5-8 in Table 3).

An example of the mean optical properties computed in the layers (versus measurement time) is given in Fig. 7 for "bu" station. The range of values taken by different variables is large. Another aspect is the lower number of values reported or retrieved for extinction and depolarization (these features are specific to all stations). The figures for all the stations will be shown in the companion paper. Systematic measurements and more intensive parameters were available for "at", "bu", "th" and "wa"

stations while over the entire analysed period, systematic measurements were provided by "at", "bu" and "th".

Once the mean optical properties were calculated, the intensive parameters were determined (where possible). All the IPs have SNR > 2. The imposed limits (data filtering) for IPs are the following: LR@355 = [20 150] sr, LR@532 = [20 150] sr, EAE = [-1 3], BAE@355/532 = [-1 3], BAE@532/1064 = [-1 3], PDR@355 = [0 0.3] and PDR@532 = [0 0.3] (following closely Burton et al., 2012; Nicolae et al., 2018). The figures representing the intensive parameters for all the stations will be shown

in the companion paper. The collected records about intensive parameters found in literature (46 reference values from 39 cited papers) are shown in Fig. S1 and Table S1 in supplement. Table S2 shows the cited references. The extreme values for LR@355 are 21 sr (Müller et al. 2005) and 130 sr (Tesche et al. 2011) while for LR@532 are 26 sr (Müller at al. 2005) and 147 sr (Mariano et al. 2010). For EAE355/532 we have 0 (Müller at al. 2005) and 2.4 (Giannakaki et al., 2016). For BAE@355/532 we found 0.35 (Tesche et al., 2011) and 2.8 (Giannakaki et al., 2010) while for BAE@532/1064 we found

0.29 (Tesche et al., 2011) and 2.85 (Gross et al., 2013). For PDR@355 the extreme values were 1% (Janicka et al., 2019) and 31% (Vaughan et al., 2018) while for PDR@532 the extreme values were 0.3% (Stachlewska et al., 2018) and 20% (Hu et al., 2018). Figure 8 shows an example (for "at") of the number of layers selected and the corresponding number of intensive parameters retrieved. One can compare with Fig. 6 over the difference between available number of optical properties and the final number of intensive parameters. An example for the intensive parameters is shown in Fig. 9 for "wa" station. For Warsaw

site, all profiles for extinction and backscatter were calculated using the classical Raman evaluation (Ansmann et al., 1992). The lines in magenta and cyan represent the minimum and the maximum as reported in literature. Several values outside the literature range are observed for EAE, both BAE and PDR@355.

The mean, median, minimum and maximum values taken by the intensive parameters for all the stations providing at least one parameter (all stations but "sf") are discussed in the companion paper. The final number of selected layers and the

corresponding time stamps are shown in Table 3, columns 9 and 10. The range of values taken by a specific parameter is large. The number of outliers dismissed based on predefined ranges of acceptable values for each intensive parameter is small (3.7% per total). For each IP we have the following numbers: 8/305 (2.6%) for LR@355nm, 8/253 (3.2%) for LR@532nm, 18/243 (7.4%) for EAE355/532, 39/642 (6.1%) for BAE355/532, 21/706 (3%) for BAE532/1064, 0/132 (0%) for PDR@355 and 0/242 (0%) for PDR@532. Please note that a thorough investigation of the outliers was not performed. Conversely, we focused



on a semi-automatic evaluation by applying different criteria for outliers' elimination (filtering). One can draw three possible reasons for the presence of outliers: a) the smoothing applied on the Raman signals, which induces a very smooth profile for extinction coefficients, b) a shift of the profile towards higher altitude (most probably due to non-accurate calibration range) and, c) a slight difference between the peaks of the backscatter versus extinction or a small difference in the slope (most probably related with the smoothing of the Raman signals). All values of the intensive parameters will be shown in the companion paper. After outliers' elimination we start the data (IP) interpretation.

## 5 Data interpretation

We have focused on several directions to investigate and interpret the measurements by means of the intensive parameters. Here we show examples for each direction, while the comprehensive analysis will be performed in the following paper.

According to Müller at al. (2005, 2007, 2016), effective radius of BB particles increases with time (distance), most probably due to coagulation and aggregation of the particles (Reid et al., 2005). On the contrary, EAE decreases with time (distance). While the heaviest particles will sediment during transport, the particles reaching measurement site are still larger than the emitted ones. It was shown that for aged BB particles LR@532 > LR@355 (e.g. Wandinger et al., 2002, Murayama et al., 2004; Müller et al., 2005; Sugimoto et al., 2010, Nicolae et al., 2013). Measurements and retrieved values of effective radius of BB particles showed that the fresh particles have a mean radius around 150 nm, while the aged particles have a radius around 300 - 400 nm (Müller et al., 2007). Particle size depends on many factors such as the combustion type in fires (in-flame or smouldering) or type of vegetation while the smoke aerosol undergoes different physical-chemical processes in atmosphere. (Reid and Hobbs, 1998). Differences between smoke properties in different regions were also found. Thus, smoke particles from Brazil absorb more and scatter less the solar radiation as compared with smoke in N America (Reid and Hobbs, 1998). Müller et al. (2005) observed differences between N America and Siberian smoke over Germany; particles from N America showed smaller size and higher EAE. Veselovskii et al. (2015) tried to relate BAE with EAE and showed that while EAE depends mainly on particle size, BAE depends both on particle size and refractive index, being very sensitive to the latter (see their Fig. 20). In specific conditions, for the region of 0.5-1.5 for EAE, BAE@532/1064 decreases (BAE@355/532 increases) with increasing EAE (see their Figs. 20 and 22). However, in different conditions for particles size, refractive index and fine mode fraction, both BAE can increase with increasing EAE (see their Fig. 19). On the other hand, Su et al. (2008) showed few scenarios for the relationship between BAE@532/1064 and EAE@553/855, which depend on RH and the contribution of the fine mode fraction. While for a high fine mode fraction in the smoke composition and RH < 85%, BAE@532/1064 increases with increasing EAE@553/855 (EAE > 2), for a dominant coarse mode and RH < 85% BAE@532/1064 decreases with increasing EAE@553/855 (EAE < 0.5) (see their Fig. 8). According to Veselovskii et al. (2018), PDR532 decreases with increasing EAE. In general, LR532 increases with increasing LR355 (e.g. Nicolae et al., 2018). Similarly, PDR532 increases with increasing PDR355 (e.g. Stachlewska et al., 2018).



Concluding the findings so far, we expect a colour ratio (CR) for LR > 1 for aged smoke (LR@355 decreases and LR@532 increases with aging smoke) and EAE decreasing with time (distance). PDR@532 increases with time (while EAE decreases). The studies showed no straightforward pattern for CR$_{BAE}$ and CR$_{PDR}$ evolution with time.

### 5.1 Fire events seen by several stations. Example for Bucharest and Thessaloniki stations.

Over the ten years period, we have found five events (with retrieved intensive parameters) recorded at two stations which have the same smoke origin (Fig. 2, stage IX in Methodology). The events belong to local fires in Europe. Two events are seen by "at" and "th" (2014 and 2017) and one event is seen by "bu" and "th" (2014), "mi" and "wa" (2015), "at" and "oh" (2016) stations. Unfortunately, we did not obtain common intensive parameters for all events. Only two events had one common intensive parameter (BAE@532/1064). One event is shortly presented below, emphasizing the methodology.

The event was recorded during 20140909-20140910 at "bu" and "th" stations (see time frame and altitude on Fig. 10 upper right plot). The backtrajectories revealed the same fire origin. Figure 11 shows the air backtrajectories for the layers detected at the two stations. The fires' location (within 100 km and +/- 1h) is shown as well. The colour and size of the fires correspond to their FRP (see legend). The Hysplit back trajectories in Fig. 11 show that the main smoke source was the fires over eastern Ukraine. Figure 11, left ("bu" station) indicates as second smoke source from the fires over eastern Romania (~ 72 h back) and

N Ukraine (~ 144 h back). The smoke particles were transported from Ukraine to Bucharest station over a period of five days descending from approximately 4200 m on 2014/09/04 to 3000 m on 2014/09/09. For the Thessaloniki station the main source was the fires over southwestern Ukraine (Fig. 11, right). The smoke particles were identified on 2014/09/10 around 2100 m. Note that for "th" case, the 'common fire' was detected at ~ 3500 m. In Fig. 10, the first two left plots we show the location of the fires which contributes to all measurements on 9 - 10$^{th}$ of September and their histogram occurrence on 1x1 grids. From

such histograms we pick the grids where we have fires contributing to both stations. See grid which has both colours, between 36 and 37 E and 47 and 48 N (also marked by a square on the upper plot). The location (longitude and latitude) of these fires from the "common" grid are shown versus fires occurrence time as well as versus measurement time at each station (Fig. 10 lower left plots). The common fire revealed through the back-trajectory occurred at 10:39 on 4$^{th}$ of September (latitude = 47.537, longitude = 36.275) and it was recorded at ~ 11:00 on 9$^{th}$ at "bu" and at ~11:00 on 10$^{st}$ of September at "th" (lower

layer left). Two layers were detected at "th" station (where one corresponds to the common fire) and one in "bu" station. We analysed if there were other fires contributing to the same measurement as in general, along backtrajectories, we encounter many fires at several locations. In the case of "bu" there were 24 other fires (detected 46 times) contributing, located in Ukraine as well as in NE Romania (occurring during 3$^{rd}$, 4$^{th}$ and 6$^{th}$ of September). For "th" we counted 35 other fires (detected 66 times), located at E Ukraine (occurring during 2$^{nd}$ and 4$^{th}$ of September). See lower plot of Fig. 10. Please note that function

of the air trajectory, one fire can be seen more than once (e.g. during a cyclone or anticyclone) or due to slow air motion (spatial-temporal stationarity over the 100 km area and 1 hour).

We can conclude that we have a mixed smoke recorded in both "bu" and "th". However, the mixtures are different for 'bu' and 'th' stations.





Figure 10 right plots shows the layers location (marked by a square in front of the middle of the layer) and intensive parameters for the layers. Note that the retrieval of the intensive parameters was not possible at all times for all layers. The two layers detected in "bu" and "th" were located at 3061 m and 2123 m respectively.

For the measurements at "bu" (20140909 10:45 UTC) and "th" (20140910 11:15 UTC) locations, which correspond to the common fire in E Ukraine (47.537 N, 36.275 E) on 20140904 10:39 (as well as other additional fires for each station), there were the following intensive parameters calculated.

"bu", 09/09 10:45, 3061 m: BAE@355/532 = 1.82 ± 0.0001, BAE@532/1064 = 1.32 ± 0.0001, PDR@532 = 6.8 ± 0.03%.

"th": 10/09 11:15, 2123 m: BAE@532/1064 = 0.51 ±0.02.

The values for BAE@532/1064 were 1.32 and 0.51 for "bu" and "th" respectively. The difference between the two values may

be explained by the different mixture of smoke (originating from different fires). All the contributing fires for "th" besides the common fire are located in E Ukraine while the contributing fires for "bu" are located in E Romania, E Ukraine and NE Ukraine. According to backtrajectories for "th", the other fires contributing to measurements are further located in time than the common fire. Lower value for BAE@532/1064 in "th" reveals a relatively larger contribution of the big particles to backscatter. On the contrary, in "bu" the smallest particles are the most efficient in backscattering.

For the other event with common IP for the same source (20170529-20150602), the smoke was labelled as of 'single fire' as no other fires were identified along the backtrajectory. This event will be discussed in the subsequent paper.

**5.2 Long-range transport from North America. Example for Athens station.**

We have identified a number of 24 events with long range transport from North America (Fig. 2, stage X in Methodology) for which we have at least one intensive parameter retrieved. The events are reported in 2009 and over 2012-2017 period. As in

previous section, for each event, we plot the Hysplit trajectory and the location of the fires along it, the histogram with the number of fires in each 1° x 1° geographical grid, the geographical location (longitude and latitude) versus occurrence time of the fires and versus measurement time at the stations(s). Last, we show the layers altitude (error bar signifies the layer' thickness) and the intensive parameters in the layers.

We have identified eight measurements periods (events) when the smoke is arriving solely from N America ("pure N

America"). The other cases represent measurements of mixed smoke, where the smoke is coming from both N America and local fires (mostly European) ("mixed"). In two cases we have mixed smoke from N America and N Africa or Middle East. Usually, during measurements period, there were also recordings of BB with other origin (Europe). For each layer we check if the smoke is coming from a single fire or more fires (count their number), and quantify the locations. We have one event when there were measurements taken at three stations and one event with measurements taken at two stations. All others cases

represent measurements recorded by a single station. As the number of intensive parameters determined for each station varies considerably, in general we cannot compare directly all IPs for the same event. As a consequence, a statistical analysis will be performed over entire set of parameters. Below we show one example of long range transport recorded in Athens which



provided more measurements and more intensive parameters. The event recorded over three days in July 2013 will be discussed in the companion paper.

### 5.2.1 Smoke event recorded on 20170713

During this day we recorded several measurements in Athens (see time and layers altitude in Fig. 12, upper right plot).

According to backtrajectory and the fire occurrence along it, we determined three layers of N America smoke origin (their location is marked with a black square at the left). As seen in the backtrajectory (Fig. 13), the fires' location in N America is different for the three layers (different source).

The first layer of smoke origin was detected at 14:27 at ~ 2900 m altitude. Our calculations show that there were different fires contributing to this measurement. Thus, we have four fires (counted eight times) from which only one fire (counted twice) was

of N America origin. This can be seen in Fig 12 left last panel as well as in the backtrajectory in Fig 13 first plot where we see the fires detected in the first two days backwards (local fires, in Greece and Italy) as well as the fire found after almost eight days back in N America. The fire from N America occurred at 20:14 on 5 July in the longitude by latitude grid [-111 -110] x [50 51]. The location of the fire is shown by blue arrows in Fig. 12.

The second layer was measured at 18:49 at 2102 m altitude. 12 fires were found (counted 19 times) to have contributed to the

measurement from which one fire (detected once) was of N America origin. See Figs. 12 and 13. The fire from N America was detected at 19:16 on 6 July in the longitude by latitude grid [-84 -83] x [42 43]. The location of the fire is shown by green arrows in Fig. 12.

The last layer was measured at 19:34 at 3872 m altitude. We found two fires (each detected once) of N America origin. The N America fires occurred at 09:47 on 6 July and were located in the longitude by latitude grid [-109 -108] x [47 48]. The location

of the fire is shown by magenta arrows in Fig. 12. The trajectory layer in Fig. 13 is the higher one (light blue). The local fires observed in the plot are detected by the first layer (at 2012 m).

Thus, we can label the first two layers as 'mixed' smoke while the third one as "pure N America" smoke. We observe that for the mixed fires, the contribution from local fires is much larger as we have 6/8 and 18/19 fires counts in Europe for first and second layer respectively. The intensive parameters for the three layers are:

• layer 1 @ 14:27 2942 m, mixed: BAE@355/532 = 1.42 ± 0.03, BAE@532/1064 = 1.39 ± 0.01
   • layer 2 @ 18:49 2102 m, mixed: LR@355 = 48.4 ± 1.04 sr, LR@532 = 33.7 ± 3.5 sr, BAE@355/532 = 1.72 ±0.05, BAE@532/1064 = 1.85 ± 0.02
   • layer 3 @ 19:34 3872 m, N America: LR@355 = 58.5 ± 2.63 sr, LR@532 = 67.2 ± 4.79 sr, EAE = 1.02 ± 0.14, BAE@355/532 = 1.36 ± 0.15, BAE@532/1064 = 2.11 ± 0.08

We observe that BAE@355/532 has the smallest value for 'pure N America' while BAE@532/1064 is the highest. Relatively small value for EAE for the third layer corresponds to bigger (coarse mode) particles, associated with aged smoke. On the other hand, values of LR (LR@355 < LR@532) suggest the presence of aged aerosol as well. The layer's altitude is the highest of all three. The values of LR for the second layer (LR@355 > LR@532) suggest fresh aerosol (of smaller size, fine mode particles). This can be supported by the contribution of the local fires (detected a few hours back). If we compare the colour



ratio (CR) of BAE we obtain values of 0.98, 1.08 and 1.55 for the three layers. $CR_{LR}$ for the second and third layers were 0.7 (fresh smoke) and 1.15 (aged aerosol). Based on all reported values in literature (Table S1), we found the following values for $CR_{LR}$ and EAE for fresh, aged and N America case (particular case of aged smoke). $CR_{LR}$ was 0.88, 1.08 and 1.23 while EAE was 1.47, 1.2, 0.95. $CR_{BAE}$ had the values 0.76, 0.98 and 0.98. We may speculate that $CR_{BAE}$ may increase with time (distance).

5 Please note that the time difference (an hour at most) between right plots and bottom left plot comes from the fact that Hysplit back-trajectories start at sharp hours. For example, for the measurement at 18:49 the starting point on back-trajectory is 18. Statistics over LRT from N America will be shown in the companion paper. We encountered 168 measurements over the 24 periods (over 2009 – 2017 period). From these measurements, 77 have a N America origin while the other 91 have a different BB origin (local). The LRT events from N America is analysed differencing between 'pure N America' fires (sensed smoke comes solely from N America) and 'mixed' fires (measured smoke is a mixture of N America smoke and local/European 10 smoke).

### 5.3 Analysis over geographical regions

#### 5.3.1. Geographical regions

Taking into account the position of the 14 stations (Fig. 1), four geographical regions are chosen as follows: SE Europe ("po", 15 "at", "th", "sf", "bu"), SW Europe ("gr", "ba" and "ev"), NE Europe ("be", "wa", "mi") and Central Europe ("ca", "le", "oh"). This corresponds to stage XI from Methodology (Fig. 2). The statistics over the measurements for each individual station show the following. Belsk and Cabauw stations focused on LRT smoke from N America with ~99 % of the measurements (1149 / 1159 and 1001 / 1013 N America fires / total number of fires). Leipzig measured 86.2 % (156/181 N America fires / total number of fires) LRT smoke. The others have measured mostly in Europe as follows (# fires in EU / # total fires): Athens 81.7 20 % (1657 / 2028), Barcelona 91 % (172 / 189), Bucharest 93.82 % (1883 / 2007 cases), Granada 68.32 % (1643 / 2405), Minsk 98.04 % (1051 / 1072), Observatory Hohenpeissenberg 74.84 % (116 / 286), Potenza 100% (142 / 142), Sofia 100% (16 / 16), Thessaloniki 87.38 % (1807 / 2068), Warsaw 85.4 % (5036 / 5897). For Evora station we encountered 44.39 % (99 /223) fires in N America and 48.43 % in Europe (108 / 223). Per regions, we have detected fires from N America as 4.5 % (282 / 6261), 8.56 % (241 / 2817), 87.55 % (1181 /1349) and 24.13 % (1961 / 8128) for SE, SW, CE and NE region respectively. The 25 number of fires detected in Europe was the following: 87.93 % (5505 / 6261), 68.26 % (1923 / 2817), 11.34 % (153 / 1349) and 75.01 % (6097 / 8128) for SE, SW, CE and NE region respectively.

In **Fig. 14** we show the location of the fires seen by the stations from SE region. Note that the grid size is 1 x 1 degree latitude and longitude. The first remarks for each cluster are the following. As mentioned, for SE region, we have a number of 282 fires located in North America (4.5 %) and 5979 elsewhere (total of 6261 fires), most of them being located in East Europe 30 (5505). The longitude region with most of the fires is between 20 E and 30 E while the latitude region is 37 - 46 N. This corresponds to the Balkan region, covering parts of Romania, Bulgaria, Macedonia, Greece. Most of the measurements were taken at "bu", "at" and "th".





### 5.3.2. Intensive parameters by geographical regions

A statistical investigation over the intensive parameters was performed for each geographical region in order to identify the main features. The analysis was performed by separating the intensive parameters based on their continental source origin. The following origin were considered: Europe (EU), Africa (AF), Asia (AS), N America (NA) and a combination of any of

them (EUAF=EU+AF, EUAS=EU+AS, EUNA=EU+NA, EUAFAS=EU+AF+AS, etc). The statistics was performed over all available cases. Here we present the results for SE region (for consistency). The other three regions will be discussed in the following paper as well as the overall assessment.

**Fig. 15** shows the scatter plots for some of the combinations between two IPs. Note that the number of pair points available for each combination is different. The following features are revealed (the mean values are discussed). In average, there is a

linear correlation between the two LR if we dismiss the value for NA. The mean LR@532 is slightly larger than the mean LR@355 ($CR_{LR} > 1$) which suggests the presence of aged aerosol in general. For EU source region, PDR@532 is below 7% (except three values) which corresponds to a low depolarization. For EAE features, we observe low value for EUAF source region (~ 0.65), while the other three source regions (EU, EUAS and EUNA) have similar values, around 1.5 (based on EAE vs BAE@355/532 plot). Large value of EAE (~1.5) suggests smaller size aerosol, specific to fresh smoke particles. The values

for the EUAS and EUNA mixtures suggest that the contribution of EU fires to the mixture is large. On the other hand, the small value for EUAF region (corresponding to bigger particles) may be due to the major contribution of the AF region (as the value is not close to the EU value) corresponding to relatively big smoke travel time. Further analysis will be performed in part II, where both $CR_{LR}$ and EAE will be accounted for when interpreting the smoke (fresh versus aged) based on the same measurements. The scatter plots between EAE and the two BAE are opposite. While EAE increases with increasing

BAE@355/532, EAE decreases with increasing BAE@532/1064 (also reported by Veselovskii et al., 2015, in special conditions). A general decreasing trend of EAE versus LR@532 is observed. The comparison between the two BAE, based on source origin, did not show a specific relationship. Based on standard deviation, we observe a big overlap among the values for all source regions. In conclusion, based on the current dataset, there is no clear separation between sources for SE region but for average values, specific trends are observed for some scatter plots.

As the dataset is limited, we cannot conclude at this stage about the existence of a clear feature with respect to the continental source origin. As mentioned on literature, EAE exhibits a decay versus time (while smoke effective radius increases with time). In the present example, for SE region we obtained quite large values for mixtures which suggest a large contribution of the European fires. On the other hand, for EUAF mixture, the value was quite small, which suggests a relatively big travel time. Medium absorption as shown by LR for both 355 nm and 532 nm (~ 50 sr) was observed for EUAS, EUAF and EU regions.

The smallest / largest value for LR@355 / LR@532 was observed for NA region, suggesting less / more absorption for small / medium size particles. The LR values for EUNA source are close to those for EU source which suggest once more the big contribution of EU fires to the mixture.





The complete analysis of IPs and CRs will be performed in the companion paper. Based on a larger IP dataset, we expect that the complete set of CR for LR, BAE and PDR, along with EAE will better characterize the measured aerosol. The statistics over all four regions will eventually bring more insights about the IPs and CRs trends versus continental source origin.

## 6. Summary and conclusions

The current study focuses on developing a methodology to analyse large amounts of biomass burning measurements by lidars. The current lidar dataset is from EARLINET database, Forest Fire category.

First, we would like to mention the current challenges when performing such analyses. Besides tackling with high amount of data, there is different data processing used by stations (using either their own algorithms or more recently SCC). The data still need quality control checks and the best way is the manual check. For more accurate results, the algorithm developed to detect

the pollution layers needs a manual check as well, and thus, the boundaries are manually corrected where required. If the investigation is based on intensive parameter results, one has to bear in mind that their number is a limited subsample of the initial data set. This is due to two main factors. First, not all of the lidar systems provide the complete sets of the intensive parameters. Second, the quality checks remove a large amount of data (more than half of the initial dataset).

The possible sources of uncertainty during such an analysis may be the following. A small change in the input to Hysplit may

give a different output (e.g. use of altitude a.g.l. versus a.s.l., use of GDAS0.5 versus GDAS1). See for example Su et al. (2015). Various algorithms employed to estimate the layer geometry may give slightly different values over the mean values in the layers. Thus, the direct comparison with other reports over the same event should be carefully performed. Uncertainties in Hysplit backtrajectories as well as in FIRMS database are not considered. Least but not last, the imperfect data quality control (including the present methodology) may contribute as well.

The current methodology (Fig. 2) describes various criteria involved in order to assure a quality-controlled dataset and the steps preceding the computation of the mean intensive parameters in the layers. The algorithm to select the layers of BB origin, employing the information from both Hysplit and FIRMS within the current criteria (*novelty*) allowed us to identify only the fires which most probably contributed to the measurements. Further, we were able to determine if the smoke measurements were originating from a single fire source or from many fire sources by identifying all the fires along backtrajectory (*novelty*).

Therefore, we could quantify the measurements as having a 'single fire' source or 'mixed fires' sources, having a N America origin ('pure N America') or a 'mixed origin' (N America and Europe). The number of fires occurring along a backtrajectory as well as the number of fires' counts was calculated (*novelty*). Further, we proposed few directions for BB study by means of the intensive parameters.

The first direction is to study the same BB event through the measurements taken at several stations. For the current dataset

we found five events as observed by two stations. The common fire source is precisely identified while the number of other fires contributing to the smoke measurement was quantified (*novelty*). For two cases, BAE@532/1064 could be compared. In the current example, the measurements represent smoke originating from several fires (mixed smoke). We found that





BAE@532/1064 had a smaller value for the station which recorded smoke transported for a longer time for the common fire. The differences between the two values can be attributed to the fact that the mixed smoke measured in the two locations originated from different fires (besides the common fire).

The second direction was the study of LRT of smoke from N America. 24 events were available. We have identified that the LRT from N America can be of N America origin only or a mixture of both N America and local fires (*novelty*). The quantification of the smoke as mixed explained various values for IPs in cases where the values were closer either to N America type or to European type. In the example shown here, we identified three layers where two were labelled as 'mixed' and one as 'pure N America'. For the later, EAE (1) and $CR_{LR} > 1$ suggested aged smoke, larger particle size. For one of the mixed smokes we had $CR_{LR} < 1$ suggesting fresh aerosol (no EAE available). This can be explained by the contribution of the local fires detected. For the 'pure N America' smoke layer we obtained the smallest BAE@355/532 and the larger value for BAE@532/1064. $CR_{BAE}$ provides the biggest value for 'pure N America' smoke.

The third direction was based on consideration of four geographical regions (SE, SW, NE and CE Europe), analysed individually. Histograms of fires' locations as detected by individual stations (through HYSPLIT and FIRMS) are presented (*novelty*), showing the predominant type of measurements taken by each station (local versus LRT).

Statistics over intensive parameters were performed in the following manner. For each geographical region, the scatter plots between various IPs were drawn and the mean values for each IP were computed function of continental BB origin. In the present example for SE region, we could observe the following (Fig. 15). LR@532 versus PDR@532 for EU source region suggest the presence of medium size particles with low depolarization and relatively high absorption. A linear dependence of LR@532 versus LR@355 was observed (as reported in literature) with colour ratios larger than one, suggesting the presence of aged smoke in average. EAE had the smaller value for EUAF source region, suggesting relatively high particle size (aged smoke). EAE values for EU, EUAS and EUNA had similar values, relatively large (~1.5) suggesting relatively small particle size (rather fresh smoke). High EAE values for mixtures (EUAS and EUNA) can be explained by the large contribution of EU fires to the mixtures. Based on SE results, we found the following trends. EAE increases with increasing BAE@355/532 and decreases with increasing BAE@532/1064. A slight EAE decrease with increasing LR@532 was observed as well. The relationship between the two BAE does not show a clear feature. Note that for the current dataset, the standard deviation is large for all the means and there is an overlap among most of the values.

One of the important outcomes of this study is the quantification (within the existing assumptions) of the fires which contribute to the smoke measurement. As observed, in most of the cases, the smoke measured have several fire sources ('mixed smoke'). Note that in the part II paper more discussion is given on the statistical analysis based on the four geographical regions considered where the data are interpreted function of continental source origin.

The current paper presents first ever such widely applied approach for the analyses of the BB optical properties derived form the lidar measurements. Full methodology was developed, from QC via selection of the layers to obtaining mean optical properties with uncertainty analyses. Comparison of optical properties of BB aerosol depending on its origin was possible and revealed interesting and distinctly different results for various regions of Europe. In general, we measure mixed smoke and





thus, we do not recommend to associate precisely a measurement with a specific source without a careful check over other possible source. Based on favourable meteorological conditions and the fires' sources locations, the analysed stations or regions provides different measurements. Thus, Central Europe mostly measures LRT smoke from N America, SW Europe mostly measure smoke originating from Iberic Peninsula and N Africa, while SE and NE Europe mostly measure smoke

originating in E Europe. The current study provides a reference for further research, including algorithm testing and aerosol typing. As for the limitations, we show that although enormous efforts are undertaken on the EARLINET-ACTRIS regular long-term observations, still the availability of the optical properties profiles in the database is limited, which is mainly due to the fact that for many stations still manual evaluation of profiles is needed. Therefore, we see a strong need for both the further development of the SCC data evaluation automated chain as well as the use of this evaluation chain by as many as possible

stations for processing of lidar data. The present methodology can reveal more insights regarding the specifics of different measurement regions versus various sources when a larger number of IPs is available.

In order to increase the number of input data and further the number of intensive parameters as well as the BB study through measurements taken at several locations, we recommend to have coordinated measurements in EARLINET during BB events. On the other hand, there are numerous events discussed in literature while the data is not reported in EARLINET yet. The data

should be treated in the same manner (use SCC). The data quality control through SCC output still needs improvement. We acknowledge the large effort put to improve SCC which is envisaged to be fully operational in a few years.

Future investigations envisage several important features to be accounted for. A more detailed analysis on grouping the sources' location using cluster analysis is envisaged, where a larger number of clusters should be chosen in order to pack more homogeneous regions with similar vegetation type. Thus, a more accurate correlation between the source type and the

measurements is envisioned. The time travel should be in some way considered. FRP will be considered to estimate the injection height and thus have more confidence that the smoke reaches the backtrajectory' altitude. The biggest challenge remains the quantification of the contribution of different fires.

*Author contributions*. MA developed the methodology and wrote the paper. All authors but MA are PIs of the stations

contributing the most to EARLINET Forest Fire category. All authors contributed to revisions of the paper.

*Competing interests*. The authors declare that they have no conflict of interest.

*Special issue statement*. This article is part of the special issue "EARLINET, the European Aerosol Research Lidar Network.

It is not associated with a conference.

Acknowledgements: *We acknowledge the use of data and imagery from LANCE FIRMS operated by the NASA/GSFC/Earth Science Data and Information System (ESDIS) with funding provided by NASA/HQ. The authors gratefully acknowledge the NOAA Air Resources Laboratory (ARL) for the provision of the HYSPLIT transport and dispersion model and/or READY*



*website (http://www.ready.noaa.gov) used in this publication. The authors acknowledge the EARLINET-ACTRIS community for provision of the aerosol lidar profiles used in this study, in particular the PIs of all stations that provided data to the Forest Fire category in the EARLINET-ACTRIS database: A. Papayannis (Athens), A. Comeron (Barcelona), A. Pietruczuk (Belsk), D. Nicolae (Bucharest), A. Apituley (Cabauw), D. Bartoli (Evora), L. Alados-Arboledas (Granada), U. Wandinger*

*(Leipzig), A. Chaikovsky (Minsk), I. Mattis (Observatory Hohenpeissenberg), A. Amodeo (Potenza), D. V. Stoyanov (Sofia), D. Balis (Thessaloniki), and I. S. Stachlewska (Warsaw). We acknowledge Wojciech Kumala, Lucja Janicka, Krzysztof Markowicz, and Rafal Fortuna (University of Warsaw) for their technical support at the ACTRIS site in Warsaw, Livio Belegante, Cristi Radu, Dragos Ene and Alexandru Dandocsi (INOE 2000) for their technical support at the ACTRIS site in Magurele.*

Funding: *The research leading to these results has received funding from the European Union Seventh Framework Programme (FP7/2007-2013) under grant agreement n° 262254, as well as the H2020 ACTRIS-2 grant n° 654109. It was also supported with following national funding: the Romanian National contracts 18N/08.02.2019 and 19PFE/17.10.2018 as well as with the European Space Agency (ESA-ESTEC) funding: The Technical assistance for Polish Radar and Lidar Mobile Observation System (POLIMOS 4000119961/16/NL/FF/mg).*

Data access: *The aerosol lidar profiles used in this study are available upon registration from EARLINET webpage* https://data.earlinet.org/earlinet/login.zul*, last access: 20191126). The FIRMS data used in the study is available upon request from* https://firms.modaps.eosdis.nasa.gov/*.*

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



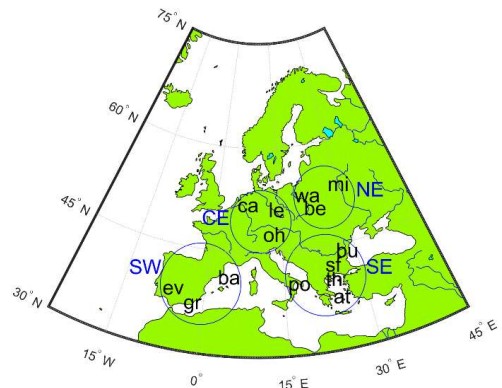

**Fig. 1. The geographical location of the 14 stations proving data for Forest Fire category in EARLINET database over 2008-2017 period. The stations are located in Athens ("at"), Barcelona ("ba"), Belsk ("be"), Bucharest ("bu"), Cabauw ("ca"), Evora ("ev"), Granada ("gr"), Leipzig ("le"), Minsk ("mi"), Observatory Hohenpeissenberg ("oh"), Potenza ("po"), Sofia ("sf"), Thessaloniki ("th") and Warsaw ("wa"). The blue circles show the four European geographical regions: South East (SE), South West (SW), North East (NE) and Central (CE).**



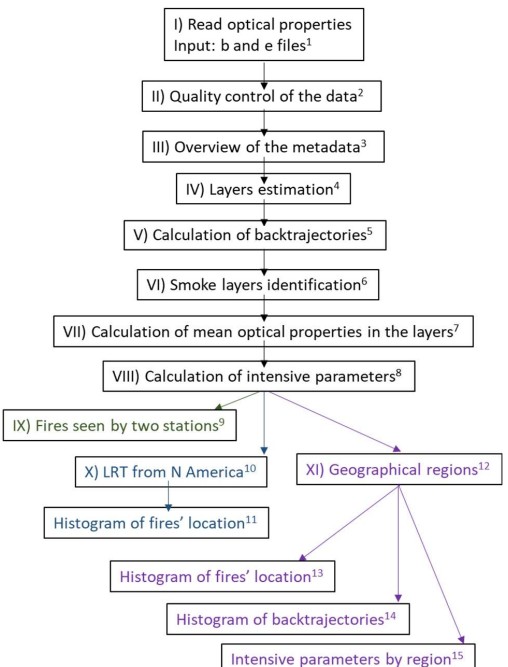

1 3589 input files; see Figs. 3, 4 and Table 1

2 EARLINET/Forest Fire files checked following EARLINET criteria; additional in-house checks

3 e.g. detection mode, evaluation method, raw resolution; see Table 2.

4 algorithm developed; non-accurate estimates were manually corrected; 1901 layers calculated for 960 time stamps

5 run Hysplit, 10 days backwards; use GDAS0.5° (23 cases on GDAS1°)

6 use Hysplit backtrajectory and fires emissions (FIRMS); 1053 layers for 677 time stamps were considered as having BB origin; see Table 3, columns 5-6.

7 the average in the layer is calculated if at least 90% data is available; the averages for which SNR<2 were dismissed; there were 1050 layers for 676 time stamps for which at least one average optical property was calculated; see Table 3, columns 7-8, e.g. Fig. 7.

8 e.g. Fig. 9; literature review on intensive parameters (Fig. S1 and Table S1). Values outside the range of acceptable values are dismissed. Remaining IPs are considered further (795 layers, 526 time stamps). See Table 4, columns 9-10.

9 based on backtrajectories, the common fires (smoke measured by two stations) were identified; compares IPs; measured smoke can be of "single fire" or "mixed fires". See Figs. 10-11.

10 based on backtrajectories, the measured smoke can be "pure N America" or "mixed" (N America + local); smoke was measured by single station or several stations; e.g. smoke measured by one station Figs. 12-13; statistics over LRT from N America to be shown in second part of the paper.

11 by station; e.g. Fig. 12.

12 region: SE ('at','bu','po','th'), NE ('be','mi','wa'), SW ('ba','ev','gr'), CE ('ca','le','oh')

13 by station; e.g. Figs. 14 for SE region.

14 main circulation pattern revealed for some regions.

15 IPs are classified based on smoke' continental origin (Europe, N America, Asia, Africa or a mixture of them). Mean values and scatter plots are considered. Fig. 15 for SE region.

Fig. 2. Methodology diagram.

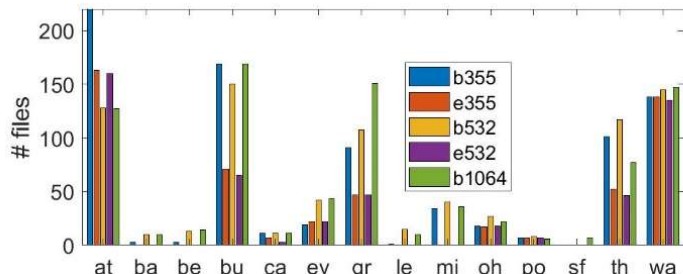

5    Fig. 3. Number of "b" and "e" files available from 14 EARLINET stations providing data for Forest Fire over 2008-2017 period.

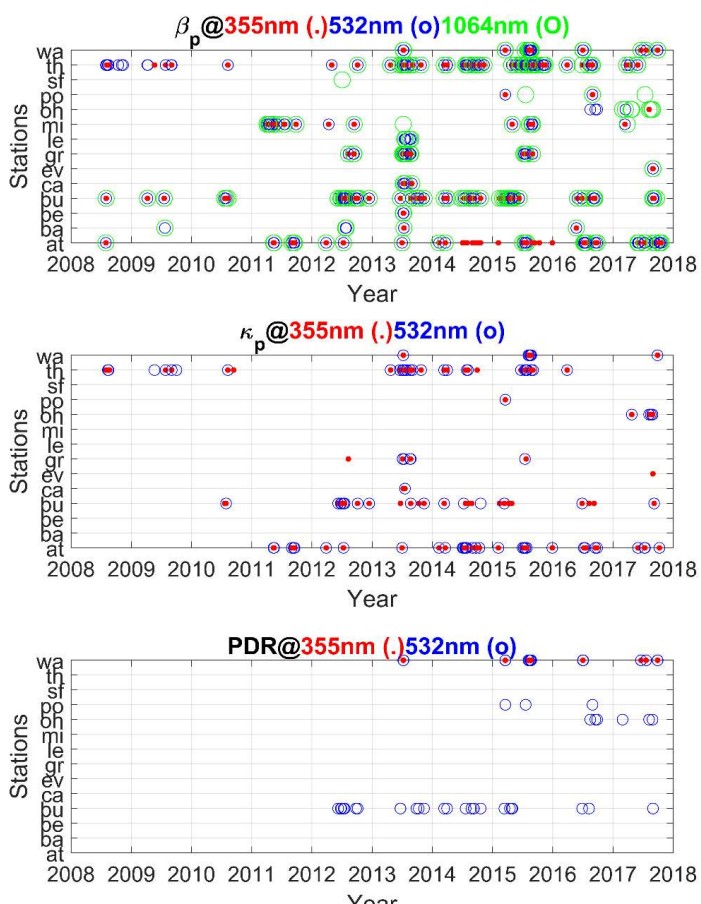

**Fig. 4.** Time span of the available particles backscatter coefficient (top), extinction coefficient (middle) and depolarization ratio (bottom) for each station, over 2008-2017 period, for the smoke layers.





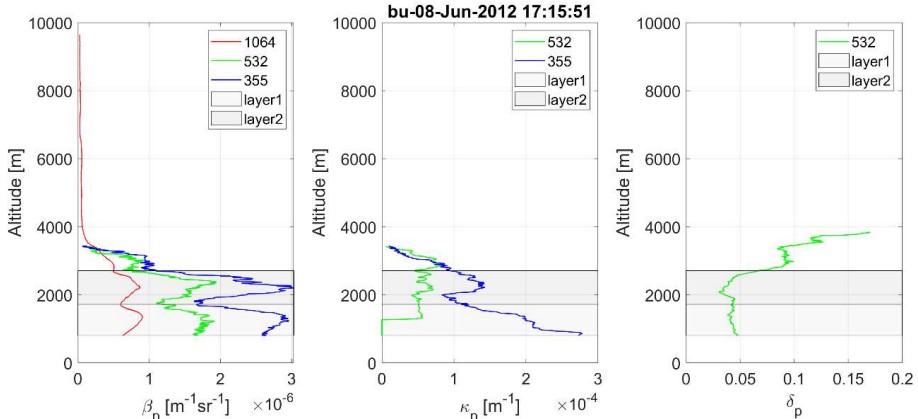

a)    Two layers automatically selected based on $\beta_{1064}$ signal. Layers' boundaries are: [802 1717] m and [1717 2707] m.

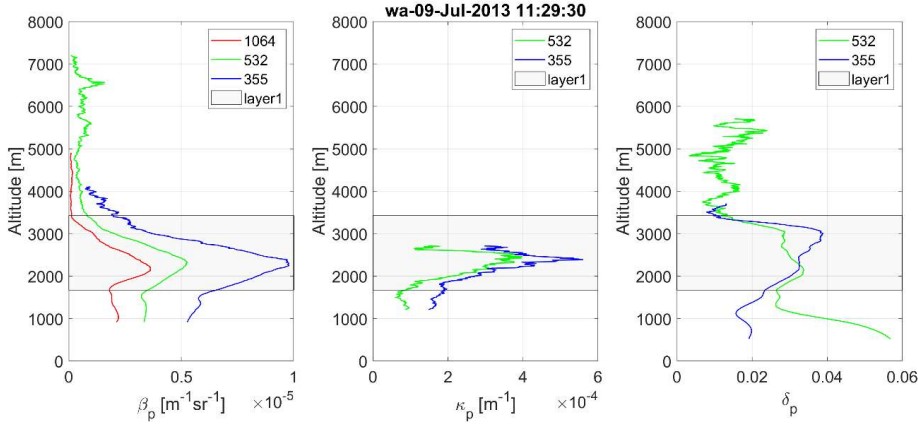

5      b)    One layer automatically selected based on $\beta_{1064}$ signal. Layer' boundaries are [1670 3426] m.





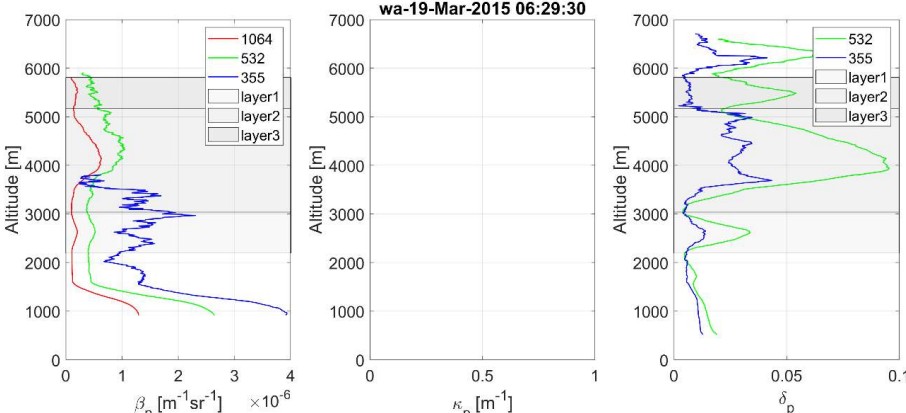

c)    Three layers automatically selected based on $\beta_{1064}$ signal. Layers' boundaries are [2193 3037] m, [3037 5166] m and [5166 5809] m.

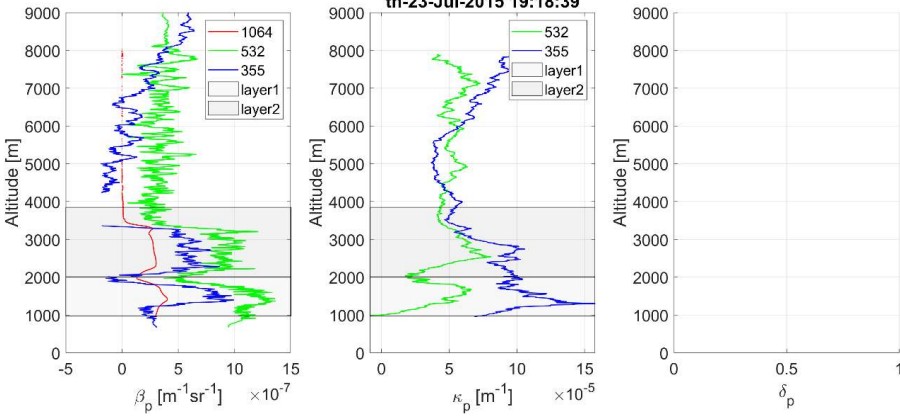

5      d)    Two layers automatically selected based on $\beta_{1064}$ signal. Layers' boundaries are [968 2002] m and [2002 3847] m.





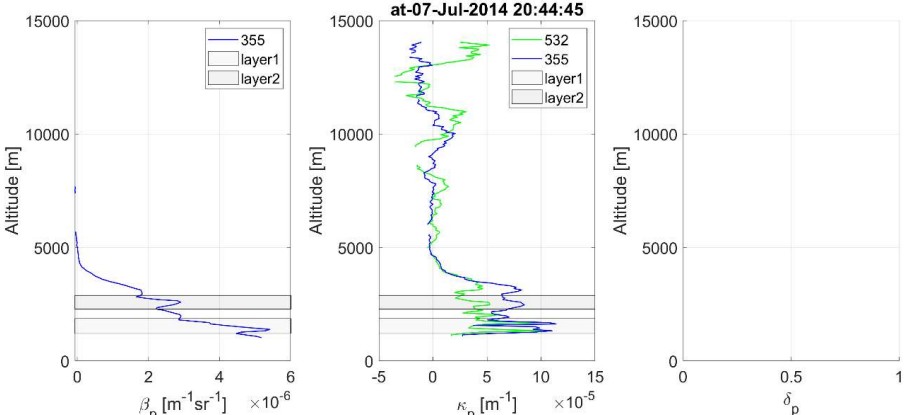

e) Two layers automatically selected, based on $\beta_{355}$ signal. Layers' boundaries are [1202 1862] m and [2282 2882] m.

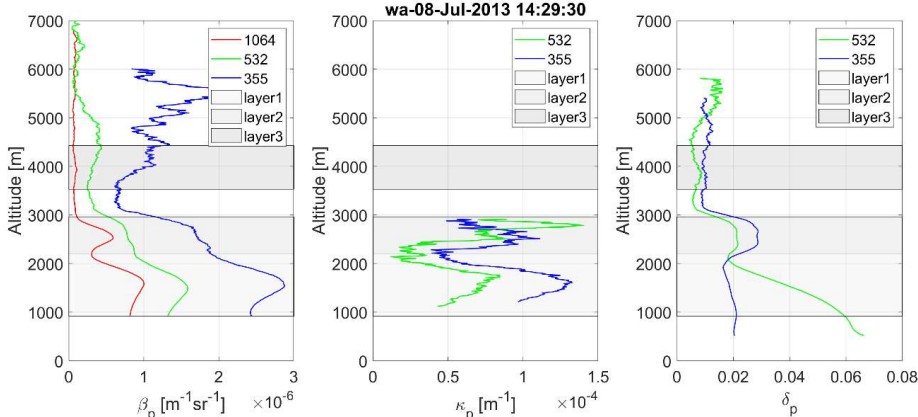

f) Three layers selected based on $\beta_{1064}$ signal. Layers' boundaries are [915 2193] m, [2193 2962] m and [3530 4427] m. The top of the second layer was manually changed from 3530 m to 2962 m. A fourth layer around 6500m was dismissed.



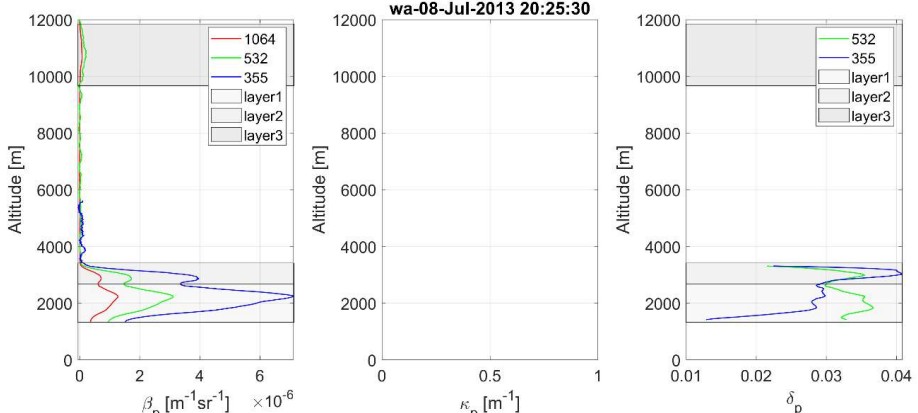

g)  Three layers selected based on $\beta_{1064}$ signal. Layers' boundaries are [1319 2678] m, [2678 3411] m and [9664 11846] m. The third layer was added manually. The top of the second layer was modified from 4061 m to 3411 m.

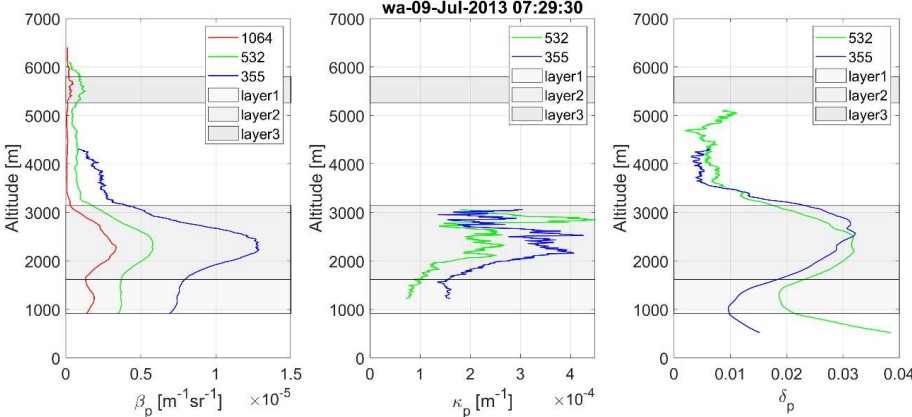

h)  Three layers detected based on $\beta_{1064}$ signal with the boundaries [915 1617] m, [1617 3142] m and [5264 5801] m. The top of the second layer was manually modified from 3612 m to 3142 m.



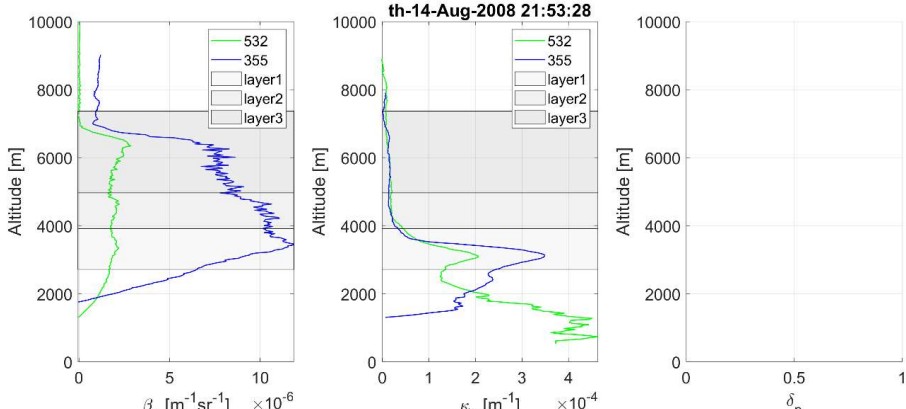

i)   Three layers selected, based on $\beta_{532}$ signal. Layers' boundaries are [2715 3915] m, [3915 4965] m and [4965 7365] m. The bottom of the second and third layers were manually changed from 4485 m to 3915m and from 5295 m to 4965m.

**Figure 5. Examples of layers selection, based on $\beta_{1064}$ signal (a-d, f-h), $\beta_{1532}$ signal (i) and $\beta_{355}$ signal (e). Layers are shown by grey areas. All available optical properties are shown (particles backscatter coefficients $\beta_p$ on the left, particles extinction coefficients $\kappa_p$ in the middle and particles linear depolarization $\delta_p$ on the right). The boundaries shown in a-e plots are the automatic output of the algorithm. In the f-i plots, one or more boundaries retrieved by the algorithm were manually adjusted.**





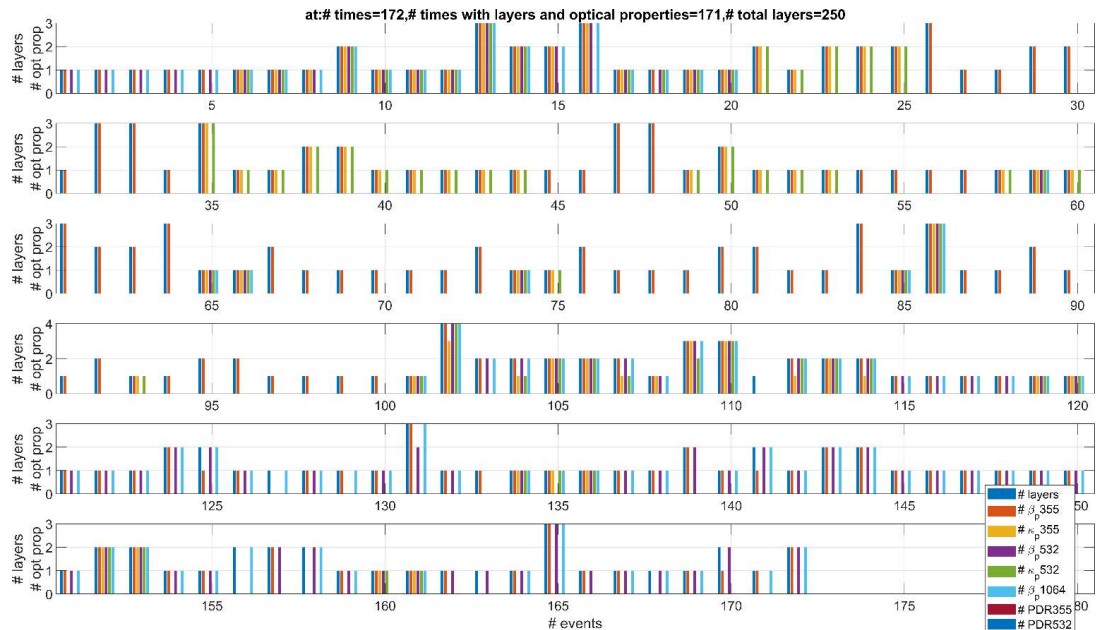

**Fig. 6. The number of times (events) when the layers were evaluated and the corresponding number of optical properties available in the layer. Example for the Athens station. Layers have a biomass burning origin (fire source). For event 111 it was not feasible to determine any optical property.**





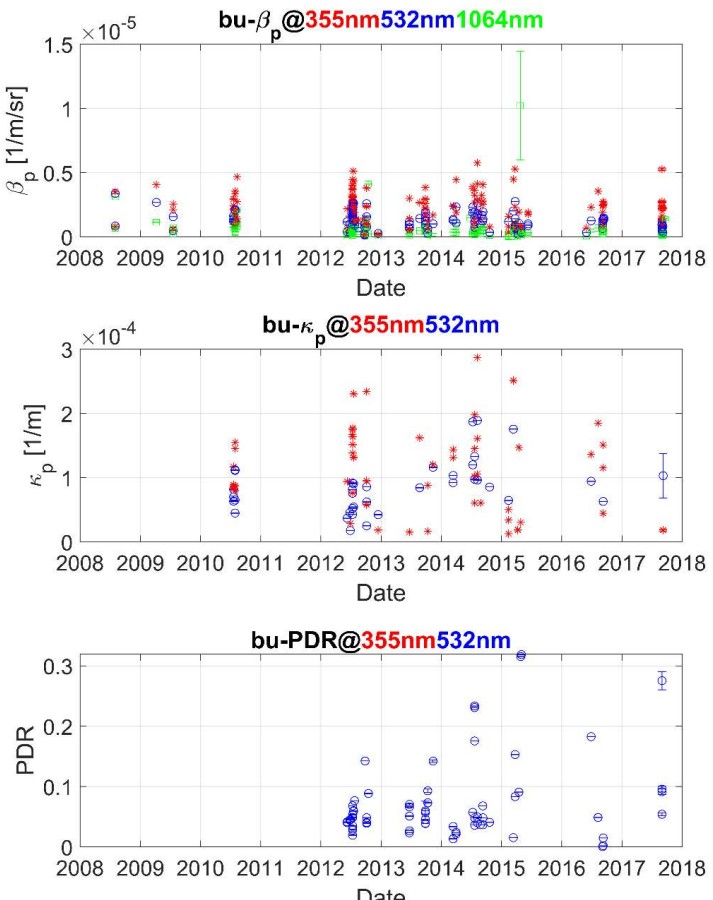

Fig. 7. Values of the mean optical properties and associated uncertainties in the layers. Example for the Bucharest ('bu') station. Layers were identified as having a fire source. Note that with few exceptions the uncertainties are very small and thus undiscernible from the mean values.



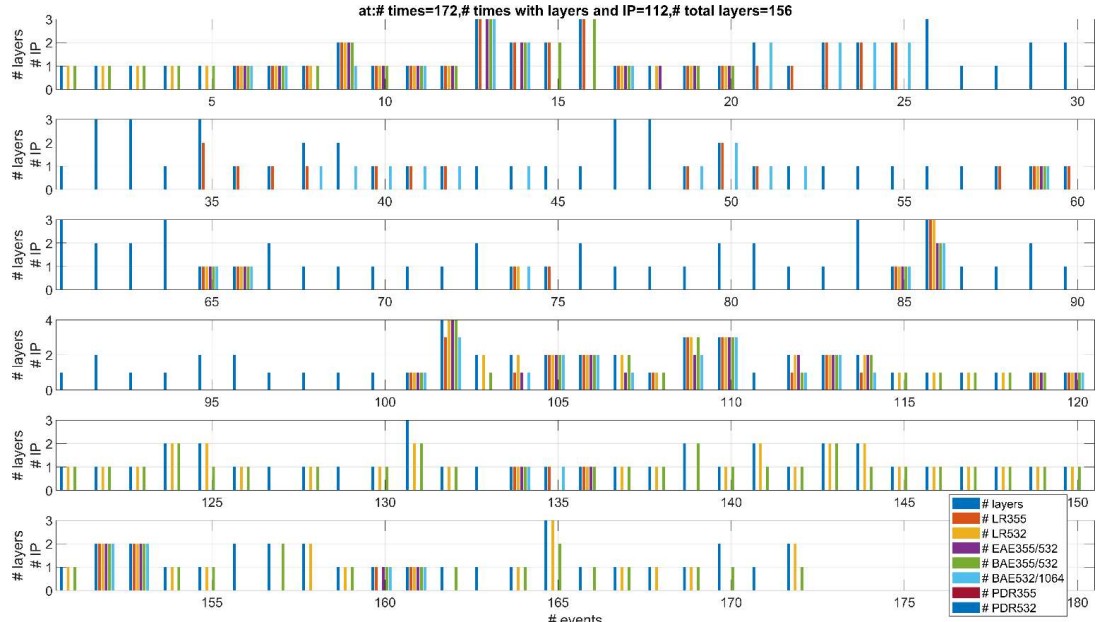

**Fig. 8.** The number of times (events) when the layers were evaluated and the corresponding number of intensive parameters available in the layer. Example for Athens station. Layers have a biomass burning origin (fire source).

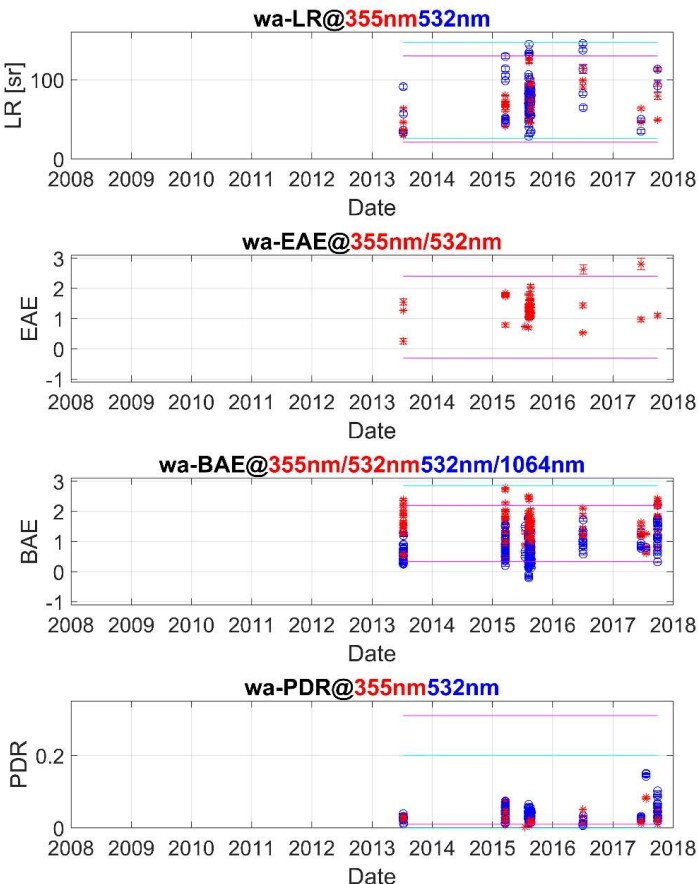

**Fig. 9. Intensive aerosol parameters derived in the layers. Example for Warsaw ("wa") station. The lines represent the minimum and maximum values reported in literature. Magenta lines represent the extreme values for the intensive parameters shown in red while the cyan lines represent the extreme values for the intensive parameters shown in blue. Layers have fire source origin.**





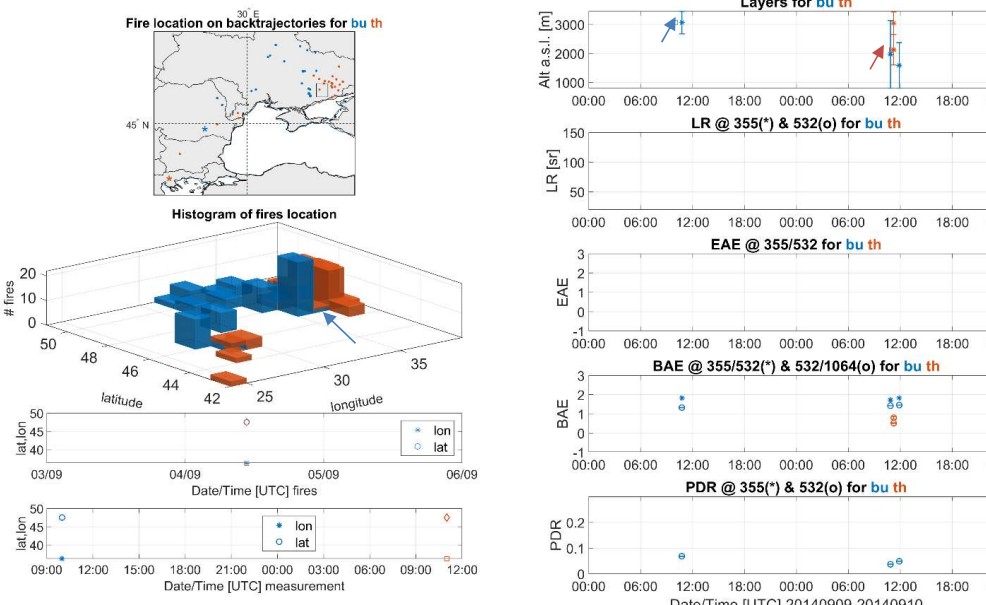

**Fig. 10. Measurements with the same source at Thessaloniki ('th) and Bucharest ('bu'). Event: 20140909-20140910. Left plots: (first) fires location seen during back-trajectories from each station (colour coded), (second) histogram of the fires occurrence in each geographical grid, (third) longitude and latitude of the fires' location versus fires' occurrence time, (forth) longitude and latitude of the fires' location versus measurement time at the two locations. Right plots: layers' altitude and intensive parameters for each station. Layers measuring the common fire are shown by arrows. The geographical location of the common fire is shown on histogram by an arrow.**





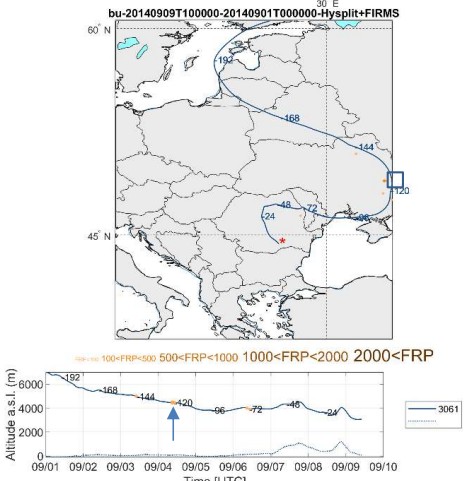 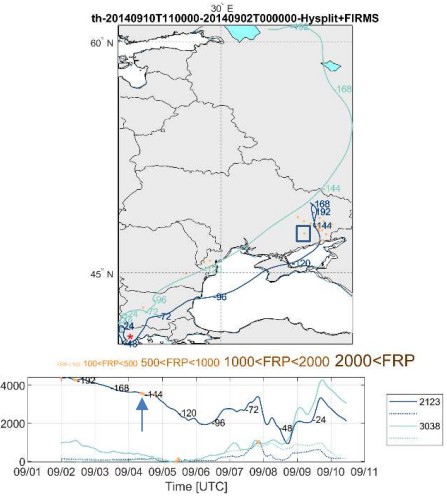

**Fig. 11. Backtrajectories and location of fires along backtrajectories within 100 km and +/- 1h for Bucharest ('bu') on 20140909 (left) and Thessaloniki ('th') on 20140910 (right). Lower plots show the altitude (a.s.l.) of the backtrajectories function of time. The fires' time is shown as well (see arrow location). The blue square denotes the geographical location of the common fire. See text for more details.**





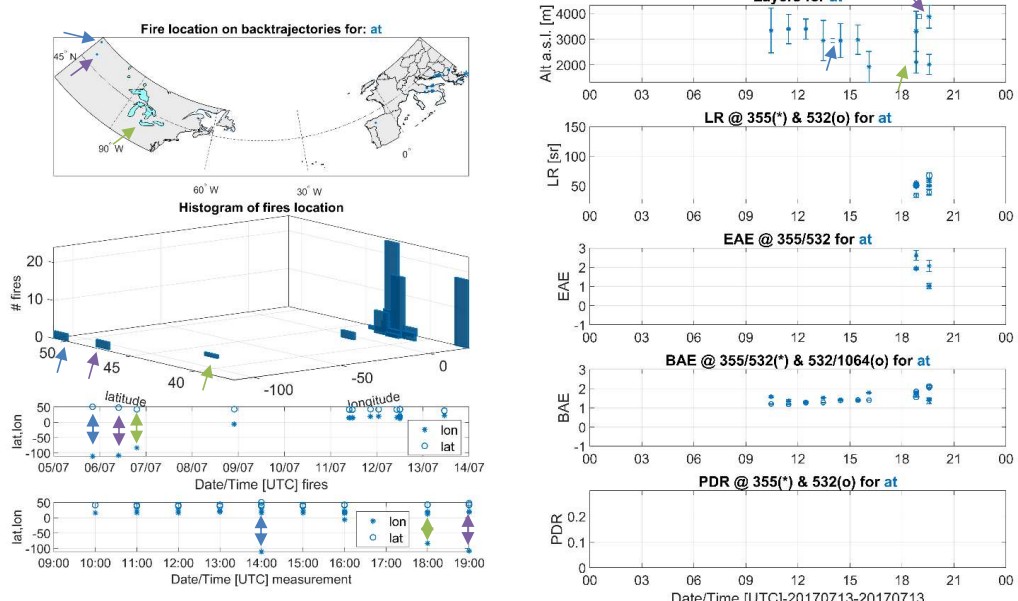

Fig. 12. LRT as measured at "at" on 20170713. First two BB layers are considered "mixed" while the third "pure NA". The arrows show the location of the fires (left plot) and the location of the smoke layers (right plot).



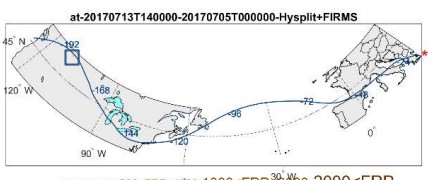

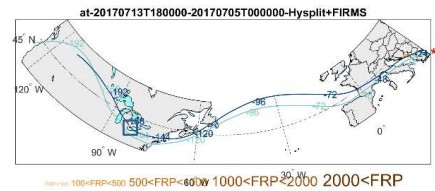

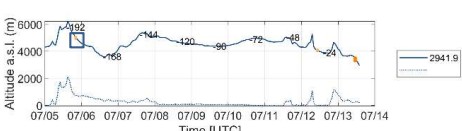

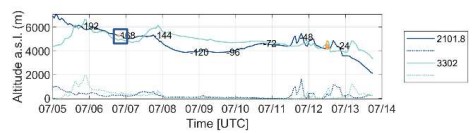

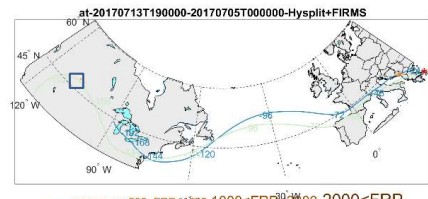

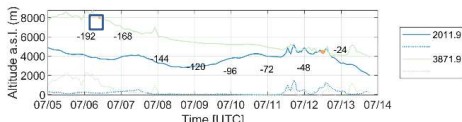

**Fig. 13. Backtrajectories for layers shown in Fig. 12 for Athens ('at') station. First two layers are considered "mixed" while the third is considered "pure N America". See text for more explanation. The squares show the location of the fire.**




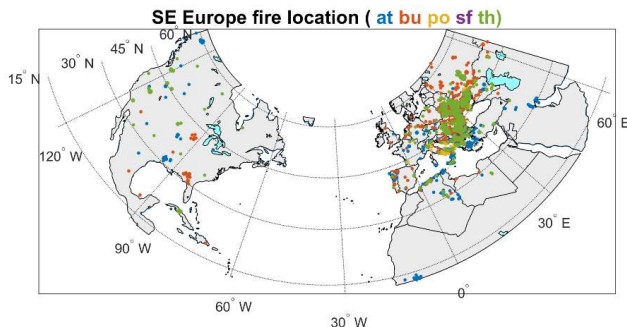

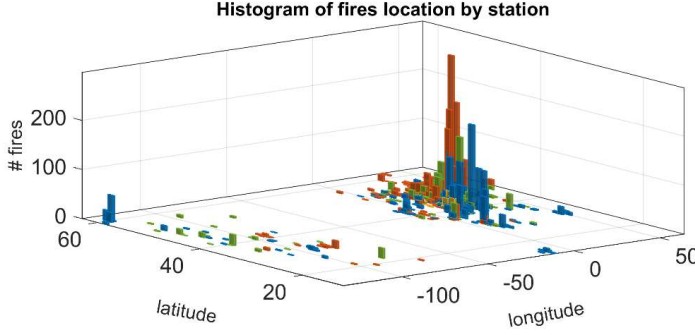

**Fig. 14.** SE Europe region formed by stations "at", "bu", "po", "sf" and "th". Upper plot shows the location of the fires detected by each station. Note that due to overlap some are not seen. The bottom plot shows the histogram of the fires detected by each station.





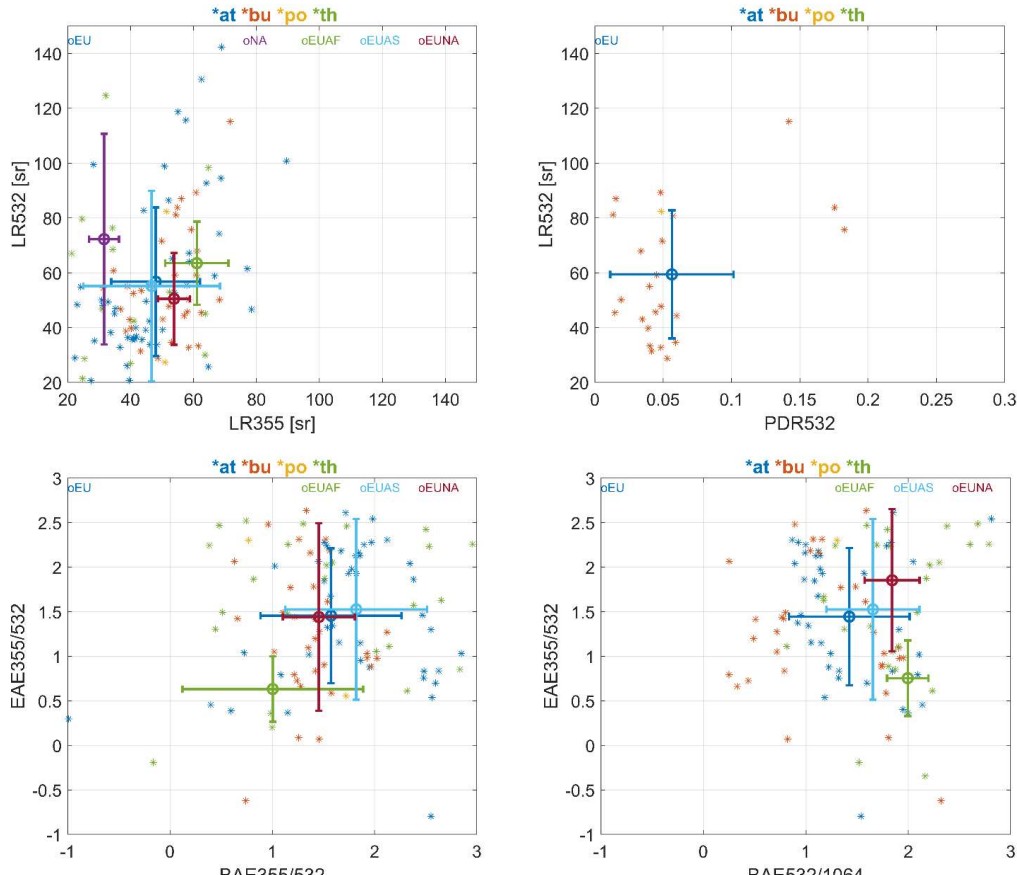





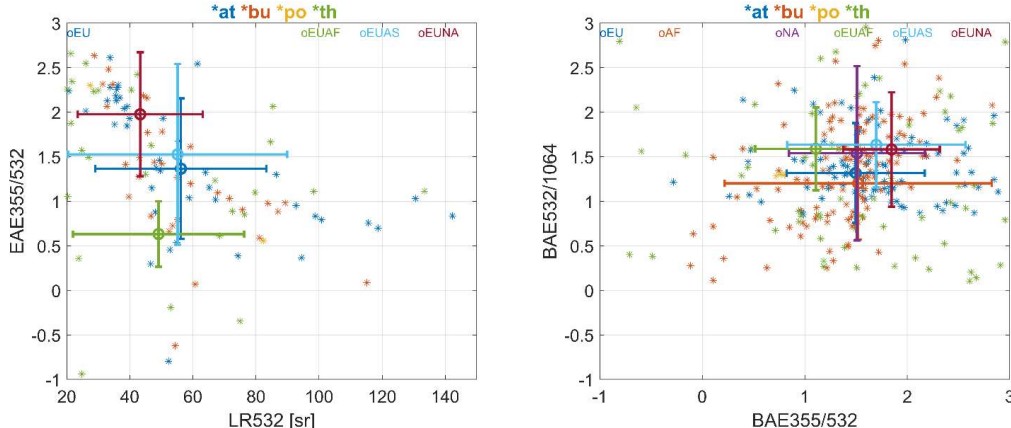

**Fig. 15 Scatter plots between various two intensive parameters for SE region. The colour code of the points is station related (as labelled in the title). The colour code for the mean and STD values is related with the source origin (as stated on the plots).**





**Table 1. List of stations and their number of backscatter (b-files) and/or extinction (e-files) at 355nm, 532nm and 1064nm. The number of particles backscatter (βp) and extinction (αp) coefficients and particles linear depolarization ratio (δp) are shown as well. Highlights systems: 3b+2e; 3b+2e+1d or 3b+2e+2d; 3b; 1b. b=backscatter, e=extinction, d=depolarization. The last column represents the number of time stamps with at least one profile of optical properties (total number being 1138).**

| | b355 files | b532 files | b1064 files | e355 files | e532 files | total b+e files | βp 355* | βp 532* | βp 1064 | αp 355 | αp 532 | δp 355 | δp 532 | # times |
|---|---|---|---|---|---|---|---|---|---|---|---|---|---|---|
| at | 324 | 128 | 127 | 163 | 160 | 902 | 324 | 128 | 127 | 163 | 160 | | | 329 |
| ba | 3 | 10 | 10 | 0 | 0 | 23 | 3 | 10 | 10 | 0 | 0 | | | 13 |
| be | 3 | 13 | 14 | 0 | 0 | 30 | 3 | 13 | 14 | 0 | 0 | | | 14 |
| bu | 169 | 150 | 169 | 71 | 65 | 624 | 169 | 153 | 169 | 71 | 65 | | 144 | 173 |
| ca** | 11 | 11 | 11 | 7 | 3 | 43 | 11 | 11 | 11 | 7 | 3 | | | 11 |
| ev | 19 | 42 | 43 | 22 | 22 | 148 | 19 | 42 | 43 | 22 | 22 | | | 43 |
| gr | 91 | 107 | 151 | 47 | 47 | 443 | 138 | 154 | 151 | 47 | 47 | | | 163 |
| le | 1 | 15 | 10 | 0 | 0 | 26 | 1 | 15 | 10 | 0 | 0 | | | 17 |
| mi | 34 | 40 | 36 | 0 | 0 | 110 | 34 | 40 | 36 | 0 | 0 | | | 42 |
| oh | 18 | 27 | 22 | 17 | 18 | 102 | 18 | 27 | 22 | 17 | 18 | | 24 | 27 |
| po | 7 | 8 | 6 | 7 | 7 | 35 | 8 | 8 | 6 | 7 | 7 | | 8 | 8 |
| sf | 0 | 0 | 7 | 0 | 0 | 7 | 0 | 0 | 7 | 0 | 0 | | | 7 |
| th | 101 | 117 | 77 | 52 | 46 | 393 | 101 | 117 | 77 | 52 | 46 | | | 140 |
| wa | 138 | 145 | 147 | 138 | 135 | 703 | 140 | 146 | 147 | 138 | 135 | 131 | 144 | 151 |
| #tot | 919 | 813 | 830 | 524 | 503 | 3589 | 969 | 864 | 830 | 524 | 503 | 131 | 320 | 4141 |
| #sta | 13 | 13 | 14 | 9 | 9 | | 12 | 12 | 13 | 8 | 8 | 1 | 4 | |

* backscatter coefficient profiles from both b-files and e-files; for the same time stamp, the profile from e-file is kept

#tot=total number of files or profiles; #sta=number of stations with particular file or profile





**Table 2. Summary of main features used to calculate the backscatter and extinction coefficient, specific for each b-files and e-files. Detection mode: 1 (photon counting), 2 (analog), 3 (analog + photon counting). Evaluation mode: 1 (Klett-Fernald), 2 (Raman), 3 (aerosol backscatter ratio).**

|  | Detection mode (1/2/3) | Evaluation method (1/2/3) | Raw resolution | Shots averaged | Zenith angle |
|---|---|---|---|---|---|
| b355 | 176/153/**590** | **524**/130/257 | **most@3.75m and 60m** | **most@1e4** | **most@0°** |
| e355 | **417**/0/107 | 0/**516**/0 | **most@60m** | **most@1e4** | **most@0°** |
| b532 | 229/199/**385** | **335**/296/174 | **most@3.75m**, then 7.5m, 15m, 60m | **most@1e4** | **most@0°** |
| e532 | **409**/0/94 | 0/**496**/0 | **most@60m** | **most@1e4** | **most@0°** |
| b1064 | 222/**608**/0 | **683**/139/0 | **most@3.75m**, then 15m and 60m | **most@1e4** | **most@0°** |





**Table 3. (2) Number of time stamps with at least one profile of optical properties (# times); (3) number of time stamps with layers and at least one optical property in the layer; (4) total number of layers for time stamps in (3); (5) number of time stamps with optical properties of fire origin. (6) total number of layers for time stamps in (5); Columns 7 and 8 represent the number of times with good optical properties and the total number of layers with good optical properties. Columns 9 and 10 represent the final number of time stamps and corresponding number of layers within imposed acceptable range.**

| 1 | 2 | 3 | 4 | 5 | 6 | 7 | 8 | 9 | 10 |
|---|---|---|---|---|---|---|---|---|---|
| Station | # times | # times with layers with optical properties | total # layers for (3) | # times with layers with optical properties of fire origin | total # layers for (5) | # times with layers with optical properties of fire origin, SNR>=2 | total # layers for (7) | # times with layers with IP of fire origin, SNR>=2, within accepted range | total # layers for (9) |
| "at" | 329 | 233 | 405 | 172 | 251 | 171 | 250 | 112 | 156 |
| "ba" | 13 | 13 | 31 | 12 | 23 | 12 | 23 | 6 | 14 |
| "be" | 14 | 14 | 58 | 14 | 21 | 14 | 21 | 13 | 20 |
| "bu" | 173 | 163 | 289 | 109 | 168 | 109 | 167 | 86 | 130 |
| "ca" | 11 | 11 | 26 | 8 | 14 | 8 | 14 | 8 | 14 |
| "ev" | 43 | 42 | 96 | 13 | 16 | 13 | 16 | 6 | 8 |
| "gr" | 163 | 146 | 296 | 80 | 120 | 80 | 120 | 68 | 97 |
| "le" | 17 | 17 | 35 | 11 | 18 | 11 | 18 | 4 | 5 |
| "mi" | 42 | 32 | 86 | 25 | 52 | 25 | 52 | 21 | 43 |
| "oh" | 27 | 21 | 50 | 12 | 18 | 12 | 17 | 11 | 15 |
| "po" | 8 | 7 | 9 | 5 | 7 | 5 | 7 | 4 | 6 |
| "sf" | 7 | 7 | 11 | 4 | 4 | 4 | 4 | 0 | 0 |
| "th" | 140 | 103 | 216 | 88 | 151 | 88 | 151 | 66 | 107 |
| "wa" | 151 | 151 | 293 | 124 | 190 | 124 | 190 | 121 | 180 |
| | 1138 | 960 | 1901 | 677 | 1053 | 676 | 1050 | 526 | 795 |

