# Peer review of "Biomass burning events measured by lidars in EARLINET. Part I. Data analysis methodology."

_Atmospheric Chemistry and Physics, 2020_

## Referee Comment (RC1) · Anonymous Referee #2 · 6 Jun 2020

Authors use database of the European Aerosol Research Lidar Network for extended period of time (2008-2017) to derive the mean optical data and intensive parameters of the forest fire products over Europe. Analysis of the lidar network observations is important but challenging task, because different lidar systems in the network have different data quality, which leads to strong scattering of retrieved parameters. Still the results provided are useful, and manuscript can be published after some revision.

Significant part of the manuscript is dedicated to description of the procedure of data treatment. No question, it is important when large volume of data from different stations is analyzed. Still this is ACP, so may be it is better to put data treatment in Appendix? But this is up to the authors.

Data quality is important and Fig.4 probably should illustrate it. However, it rises a lot

of questions. For example, in Fig.4a, extinction. in upper layer ( 2000 m) at 532 nm is stable, but at 355 nm it oscillates. Is it real or just artifact? On Fig.4b the peak of extinction is more narrow than that of backscattering. Why? At 3000 m depolarization at 355 nm becomes larger than at 532 nm. Is it real? In Fig.4c,d when backscattering coefficients at355 nm are calculated, the reference points are not shown and it is not clear, if these exist. Actually, every plot provides a lot of questions and reader will definitely be confused. The uncertainties should be provided to separate real results from artifacts.

---

## Referee Comment (RC2) · Anonymous Referee #3 · 22 Jul 2020

Review of Biomass burning events measured by lidars in EARLINET. Part I.by M. Adam et al.

This paper describes, how a long-term data set from different lidar systems has been created and combined with hysplit backtrajectories and a mask for forest fires, to obtain a tool to analyse biomass burning aerosol via remote sensing.

This is a very important and relevant topic. I fully understand that, as the authors claim, such a large, diverse data set does not run fully automatically through any software, but that user interaction is still required to create the final product. Hence, I assume that lots of work went into this manuscript. However, in the current form I still see room for improvements, both from technical and conceptual aspects.

Let me start from the principal point of view. For a publication in ACP the described method should be so clear, novel and comprehensive that the same methodology could be applied elsewhere and stands out as an independent result. Reading the manuscript in the current form I was wondering, whether a publication in a data journal with a sound description of according metadata would be an alternative. In any case the authors may consider to publish the data in itself. My feeling is that this will be worth it (see below).

From a technical point of view the error estimation should be improved. In the manuscript (see my detailed remarks below) the authors state to have a SNR of 2 as a limit and derive astonishingly small error intervals for intensive quantities. To me this does not fit together. I am convinced that a trustful evaluation of backscatter with SNR >=2 is on the edge, but the derivation of extinction or intensive quantities will be impossible. Maybe the SNR > 2 refers to raw data of individual shots, but the final analysis is done with a coarser resolution? Please make this clear. One possible usage of this paper might be by researchers outside the lidar community for further analysis and then a trustful, transparent error analysis will be mandatory.

Further, there is hardly any description of underlying lidar data from the EARLINET data base. I was wondering, looking at BAE in Fig 15, whether all lidar stations are using the same boundary condition to assess the backscatter?

To me, one of the overarching questions (not necessarily to be answered by this paper alone) is "to what extent the information content of lidar data is sufficient to track changes in the microphysical properties of (BB) aerosol". I assume that the existing data set, presented in this work, is the best and most complete from quality-assured ground based stations we have. Further immediate applications would be to not only apply a fire mask but generally an aerosol emission map to further distinction between pure BB from mixed events. Also, soil and vegetation maps may be included (In the future). A statement that a fire occurred in country XY at coordinates zz can only be the starting point for a deeper analysis. At the end, the question will be what type of vegetation on what kind of ground is burning at a shortage or not of oxygen in dry or

moist conditions. These factors will probably later on determine the values of the intensive aerosol parameters. Such a work can clearly not be done by the authors here but shows the value of their data set.

In section 5 the underlying hypothesis could be sharpened, e.g. by stating that with time / travel distance / max. encountered humidity the AEA, BAE, PDR may change like that . . . and for this reason the North American events may stay out like . . . or in the scatter plots the quantity xxx is plotted vs. yyy

So my opinion on this paper is mixed: provided that the error calculation is revised and honestly be stressed what can or cannot done with this data set, the scientific significance of the data set is very high. In its current form the papers is lacking substantial novel conclusions, though. Due to the principal importance of this data set and the work the authors performed so far to create it I rate the scientific significance as "excellent" (provided the error treatment will be revised). Due to the overall quality of the paper in its current form I rate the scientific quality as "low".

Some specific remarks are:

Intro line 19: what do you mean be fuel with respect to climate change? Fig 2, No.8 what are the limits for neglecting intensive parameters, what percentage of data has been rejected?

Section 4 P 7, l 13: with kappa you mean the extinction, right? Before you have used the symbol "e" L 20: is there a reason for the bin numbers? (the product of resolution * numbers is not constant) how does this different resolution later affects you layer selection? P8, L 4: I do not understand the possibility that a maximum is not surrounded by 2 minima. findpeaks is employed for beta1064, i.e. real data, so I assume intervals with constant values are at least not frequent? P8, L 7: in which way this last criterion is independent from the others (and hence needed)? L24/25: "Our approach . . . in line with . . ." this statement is completely unproven. Maybe you do not need it but to prove it you should apply your code to one of the examples of the

mentioned papers. P9 L 15 and 16: where is table S2? In my version of the manuscript the citations are not numbered. P10, L 24 I do not understand the definition of data availability: Isn't the ratio between layer thickness and resolution simply the number of data points? Did you use an additional criterion to flag a given data point as valid? P11, L 11 and following: is SNR >=2 really sufficient to determine particle extinction and lidar ratio? The SNR of the underlying Raman channel must have been much better, I assume. Fig 9 is interesting. However, presented like this: almost spherical particles, very high LR and very small particles I am wondering whether this is really BB or anthropogenic pollution. It would be interesting to show the hysplit trajectories for the extreme cases (maybe 2015, 2016, 2017). P13 L2: Sorry, I missed that: why PDR (at 532) increases with time? I would have assumed that coagulation makes the particle shape more spherical

P13 L 6 (and P14 later) I would omit mentioning Fig 2 for things which the reader cannot verify on his own. P14, L 7-8 the errors in the IP are unbelievable small and non-proven. If this is the error of many different measurements, you rely on the fact that the aerosol layer did not change during the sampling. Contrary, the different BAE between "bu" and "th" are much more understandable, if the fire conditions, the heat, the burning plants the amount of water vapour . . . etc changed.

---

## Author Response (AR1)

[revised manuscript text omitted]

**Anonymous Referee #2**

We would like to thank the reviewer for the thorough revision of the manuscript and for suggestions! We hope we addressed the comments accordingly.

We use across the text (below) the following highlights:

In red, reviewer's comments.

In black, our comments.

In green, the citations from the manuscript submitted.

In blue, the changed text for the revised manuscript.

Before answering to the reviewer's questions/comments etc, we would like to indicate a few changes that we have done to the initial manuscript. Those changes are not related with the data processing and analysing or its scientific content but rather with text cosmetics.

- added Faculty of Physics for co-author 2)

- in Acknowledgements we added Dominika Szczepanik

- in Funding, we added:

*The research was partially funded by the European Regional Development Fund through the Competitiveness 613 Operational Programme 2014-2020, POC-A.1-A.1.1.1- F- 2015, project Research Centre for Environment and Earth 614 Observation CEO-Terra, SMIS code 108109, contract No. 152/2016.*

- in References we added:

Adam, M., Nicolae, D., Belegante, L., Stachlewska, I. S., Janicka, L., Szczepanik, D., Mylonaki, M., Papanikolaou, C. A., Siomos, N., Voudouri, K. A., Alados-Arboledas, L., Bravo-Aranda, J. A., Apituley, A., Papagiannopoulos, N., Mona, L., Mattis, I., Chaikovsky, A., Sicard, M., Muñoz-Porcar, C., Pietruczuk, A., Bortoli, D., Baars, H., Grigorov, I., and Peshev, Z.: Biomass burning events measured by lidars in EARLINET. Part II. Results and discussions, Atmos. Chem. Phys. Discuss., https://doi.org/10.5194/acp-2020-647, in review, 2020.

Janicka, L. and Stachlewska, I. S.: Properties of biomass burning aerosol mixtures derived at fine temporal and spatial scales from Raman lidar measurements: Part I optical properties, Atmos. Chem. Phys. Discuss., https://doi.org/10.5194/acp-2019-207, 2019.

Müller, D., Ansmann, A., Mattis, I., Tesche, M., Wandinger, U., Althausen, D., and Pisani, G.: Aerosol-type-dependent lidar ratios observed with Raman lidar, J. Geophys. Res., 112, D16202, doi:10.1029/2006JD008292, 2007.

- Based on suggestions from Earlinet community, we changed the stations acronyms from two letters to three letters, according to the new nomenclature. This was already implemented in Part II. Thus, all the figures with references at station acronyms (on labels or title) were changed for clarity and consistency. Similar holds for tables.

- based on editor's suggestion for Part II, we changed N to North (for N America and N Africa)

- based on suggestions from the editor of Part II, we refer to the stations in the text by the full name instead of using acronyms.

- EAE355/532 was changed to EAE in the text (two instances: pp 11, line 18 and line34).

- a list with the acronyms used in the text was added in Supplement (as in Part II) for consistency. The reference at the acronyms list is mentioned at the end of Introduction.

A list of acronyms used in the current work is given in the Supplement (Table S1).

- figs. 10, 11, 12, 13 and 14 (now 7, 8, 9, 10 and 11): we added a), b) etc for each plot for an easier reference. We made small changes to the text and changed the3 figures caption:

Fig. 10 upper right plot -> Fig. 7 a

5   In Fig. 10, the first two left plots we show -> in Fig. 7 a-b we show

Fig. 10 lower left plots -> Fig. 7 c-d

Figure 7 caption:

Fig. 10. Measurements with the same source at Thessaloniki ('th') and Bucharest ('bu'). Event: 20140909-20140910. Left plots: (first) fires location seen during back-trajectories from each station (colour coded), (second) histogram of the fires occurrence
10   in each geographical grid, (third) longitude and latitude of the fires' location versus fires' occurrence time, (forth) longitude and latitude of the fires' location versus measurement time at the two locations. Right plots: layers' altitude and intensive parameters for each station. Layers measuring the common fire are shown by arrows. The geographical location of the common fire is shown on histogram by an arrow.

Figure 7. Measurements with the same source in Thessaloniki ("the") and Bucharest ("ino") during 20140909-20140910. a)
15   Fires location seen during back-trajectories from each station (colour coded); b) Histogram of the fires occurrence in each geographical grid; c) Longitude and latitude of the fires' location versus fires' occurrence time; d) Longitude and latitude of the fires' location versus measurement time at the two locations; e–i) layers altitude and intensive parameters for each station. Layers measuring the common fire are shown by arrows. The geographical location of the common fire is shown on histogram by an arrow.

20   Figure 11, left -> Figure 8 a)

Fig. 11, right -> Fig. 8 b)

See lower plot of Fig. 10 -> See Fig. 8 b)

Figure 8 caption:

Fig. 11. Backtrajectories and location of fires along backtrajectories within 100 km and +/- 1h for Bucharest ('bu') on 20140909
25   (left) and Thessaloniki ('th') on 20140910 (right). Lower plots show the altitude (a.s.l.) of the backtrajectories function of time. The fires' time is shown as well (see arrow location). The blue square denotes the geographical location of the common fire. See text for more details.

Figure 8. Backtrajectories and location of fires along backtrajectories within 100 km and +/- 1h for Bucharest ("ino") on 20140909 (a) and Thessaloniki ("the") on 20140910 (b). Lower plots show the altitude (a.s.l.) of the backtrajectories versus
30   time. The common fire location is marked with an arrow. See text for more details.

Fig. 12, upper right plot -> Fig. 9 e

blue arrows in Fig. 12 -> blue arrows in Fig. 9 (a-e)

green arrows in Fig. 12 -> green arrows in Fig. 9 (a-e)

magenta arrows in Fig. 12 -> magenta arrows in Fig. 9 (a-e)

35   Figure 9 caption:

Fig. 12. LRT as measured at "at" on 20170713. First two BB layers are considered "mixed" while the third "pure NA". The arrows show the location of the fires (left plot) and the location of the smoke layers (right plot).

Figure 9. LRT as measured over Athens ("atz") on 20170713. a) Location of the fires; b) Histogram of the fires. The North America fires are marked by arrows; c) Fires' coordinates versus fires' occurrence time; d) Fires' coordinates versus

measurements time; e) Location of the layers marked by arrows. First two BB layers are considered "mixed" while the third (magenta) "pure NA"; f) – i) Intensive parameters.

The trajectory layer in Fig. 13 is the higher one (light blue). -> The trajectory layer in Fig. 10 c) is the higher one (light blue).

Figure 10 caption:

Fig. 13. Backtrajectories for layers shown in Fig. 12 for Athens ('at') station. First two layers are considered "mixed" while the third is considered "pure N America". See text for more explanation. The squares show the location of the fire.

Figure 10. Backtrajectories for layers shown in Fig. 9 for Athens ("atz") station. Layers in a) and b) are considered "mixed" while the layer in c) is considered "pure N America". See text for more explanation. The fire location is marked with an arrow.

Figure 11…. Upper plot shows the location of the fires…-> Figure 11… a) Location of the fires

Figure 11… The bottom plot shows the histogram of the fires -> Figure 11… b) Histogram of the fires

Figure 11 caption:

Fig. 14. SE Europe region formed by stations "at", "bu", "po", "sf" and "th". Upper plot shows the location of the fires detected by each station. Note that due to overlap some are not seen. The bottom plot shows the histogram of the fires detected by each station.

Figure 11. SE Europe region formed by stations Athens ("atz"), Bucharest ("ino"), Potenza ("pot"), Sofia ("sof") and Thessaloniki ("the"). a) Location of the fires detected by each station. Note that due to overlap some are not seen. b) Histogram of the fires detected by each station.

Figure 12 caption:

Fig. 15 Scatter plots between various two intensive parameters for SE region. The colour code of the points is station related (as labelled in the title). The colour code for the mean and STD values is related with the source origin (as stated on the plots).

Figure 12. Scatter plots between various two intensive parameters for SE region (LR@532 vs LR@355, LR@532 vs PDR@532, EAE355/532 vs BAE355/532, EAE355/532 vs BAE532/1064, EAE355/532 vs LR@532 and BAE532/1064 vs BAE355/532). The colour code of the points is station related (as labelled in the title). The colour code for the mean and STD values is related with the source origin (as stated on the plots).

Pp 4, line 18, pp 6, line 22: change 60 % to 65 %.

- Based on editor suggestion for part II, we do not discuss any more in Part II the other event for "common fire" analysis (here section 5.1). Thus, we will add few comments here at the end of section 5.1. Note that it was an error on the manuscript: we mistakenly wrote 20150602 instead of 20170602.

Initial:

„For the other event with common IP for the same source (20170529-20150602), the smoke was labelled as of 'single fire' as no other fires were identified along the backtrajectory. This event will be discussed in the subsequent paper."

Changed:

For the other event with common IP, recorded in Athens and Thessaloniki during 20170529-20170602, the smoke was labelled as of 'single fire' as no other fires were identified along the backtrajectory. The common fire occurred on 26th of May at midnight in Ukraine (48.171 N, 30.622 E) and it was recorded in Thessaloniki and Athens on 29/05 and 31/05 respectively. BAE@532/1064 value in Thessaloniki was less than half of that in Athens, while BAE@355/532 was larger for Thessaloniki. High BAE corresponds to higher backscatter at smaller wavelengths, which indicates a higher number of small particles. The values in Thessaloniki correspond to a higher number of small size particles (at 355 nm) and with a higher proportion of large particles (at 1064 nm) compared to the ones over Athens. $CR_{BAE}$ (colour ratio of the backscatter Ångström exponents) increases from Thessaloniki to Athens (0.22 to 0.78, respectively), which suggests an increase with travel distance (time). As $CR_{LR}$

(colour ratio of the lidar ratios) and EAE (extinction Ångström exponent) were not available to characterize the smoke in terms of age, we classified the smoke as aged based on the duration of the travel time.

Overall, we conclude that the number of common events as well as the number of the common IPs is limited and, thus, no thorough examination of these events is possible. The most important feature of this analysis is that it enables us to quantify the smoke as of 'single fire' or 'mixed' and hence explain various IP values. This kind of analysis can be successfully applied in the future, when more data become available.

**Answers to specific comments of the Referee:**

Significant part of the manuscript is dedicated to description of the procedure of data treatment. No question, it is important when large volume of data from different stations is analyzed. Still this is ACP, so may be it is better to put data treatment in Appendix? But this is up to the authors.

Thank you for this suggestion. We moved to Appendix Figs. 5, 6 and 8 as well as Table 2 (now they cite as Figs. S1, S2 and S4, Table S2). The figures and tables were re-numbered. We consider that Chapter 3 on data quality control is not large and thus we would like to keep it in the main manuscript. As different criteria involved in QC are discussed along various steps of the procedure, it is difficult to move sparse parts to Appendix. We moved to Appendix the description of the algorithm to determine the aerosol boundary layer (Section 3 in Supplement).

Data quality is important and Fig.4 probably should illustrate it. However, it rises a lot of questions.

Actually, every plot provides a lot of questions and reader will definitely be confused. The uncertainties should be provided to separate real results from artifacts.

Regarding the examples in Fig. 5 (not 4 as mentioned), we added the uncertainties as suggested. In general, we cannot comment precisely on the accuracy of the optical properties profiles as related to different factors. In the database of Earlinet such information is not provided and therefore we could not investigate how they originated (e.g. calibration region, depolarization constant etc). **The input data in the study were the b-files and e-files containing the optical parameters and associated errors (which were quality checked by Earlinet QC tools and approved by the PIs of the stations).** However, a series of additional quality checks were implemented within our study (discussed in detail in the text). We performed an investigation about the profiles shown in Fig. 5 to show how we managed the IPs values.

For example, in Fig.4a, extinction. in upper layer ( 2000 m) at 532 nm is stable, but at 355 nm it oscillates. Is it real or just artifact?

**Fig. 5a).** In fact, this illustrated the special means of additional quality check that we conducted in our study. Note that large uncertainties were seen for backscatter at 532 nm above ~ 3km (x-axis not shown at full scale on left plot) and for extinction at 532 nm in the first range bins. Further, depolarization is shown as zero for few hundred meters, which is an artefact. We cannot say for sure if extinction at 532 nm has an artefact but likely 532 channel had some problems at this time. Therefore, we checked if the final IPs data set contains data from this measurement. There is no IP associated with this time stamp. The reason for this is that for these layers we did not detect any fire along backtrajectories and thus the data was dismissed (before any other quality checks).

On Fig.4b the peak of extinction is more narrow than that of backscattering. Why? At 3000 m depolarization at 355 nm becomes larger than at 532 nm. Is it real?

**Fig. 5b).** Extinction profiles for 355 and 532 are extracted from e-file (as mentioned in the text). For this particular case, there is 101 bins 'smooth running' (49 bins are used in b-files). Evaluation method is Raman in both b-files and e-files. Regarding the peaks, the extinction profiles may present artefacts towards last validated bins. The data were eliminated above 2.7 km (as provided in e-file). In the paper by Ortiz-Amezscua et al. (2017), the authors show profiles of backscatter and extinction as well as LR and PDR (their Fig. 8) for 00:00-01:00 interval. Their extinction profiles go up to ~3.5 km. Unfortunately, they do not report PDR@355nm. PDR@532 is ~ 3% (similar with the present plot, but note that their layer is estimated differently). We did not find a fire for this case either, so the data was eliminated from analyses. Most probably, extinction at 532 nm would

have been eliminated when computing the mean values in the layer as we wouldn't have had 90 % of the data available. Looking at closer measurements in our dataset, we found the following PDR mean values in layers as: 23:28 on 08/07 PDR@355=2.96 and PDR@532=3.01, 06:29 on 09/07 PDR@355=2.84 and PDR@532=3.17, 15:22 on 09/07 PDR@355=2.77 and PDR@532=2.87. As seen, the values are very close in value. However, there are regions where PDR@355 > PDR@532.

5  This is possible, and it is reported in literature e.g. Janicka et al. 2017, Harrig et al. 2019, Baars et al. 2019.

In Fig.4c,d when backscattering coefficients at355 nm are calculated, the reference points are not shown and it is not clear, if these exist.

**Fig. 5c)-d).** We did not investigate the reference points (nor showed on plots). The reference points are not mandatory variable in the Earlinet database and they are sporadically reported.

10  For **Fig. 5c)** the data processing was performed with in-house (PollyXT) algorithm and we have the following information: @355 'Ref.value 4 50 1/m*sr at 3500.5m', @532 'Ref.value 1 70 1/m*sr at 7352m', @1064 'Ref.value 6 0 1/m*sr at 8300.5m' (all using Photon Counting and Raman as evaluation method). We obtained the following IPs for the three layers (all with BB origin): 2.7 for BAE@355/532 (Ist layer), 2.6, 1.1 and 1.5 for BAE@532/1064, and 2.1, 5.6 and 4.1 % for PDR@532. Due to various criteria, the values for PDR@355 were not estimated.

15  For the plot in **Fig. 5d)** the following information is given in the file (processed with SCC): @b355 'find calibr. interval (width = 500m) between 5000 and 9000m with method: min. of sig ratio; bsc. ratio = 1.0E+000' (backscatter ratio method), @b532 'find calibr. interval (width = 500m) between 3000 and 9000m with method: min. of sig ratio; bsc. ratio = 1.2E+000' (backscatter ratio method), @b1064 'find calibr. interval (width = 500m) between 2000 and 5000m with method: min. of sig ratio; ' (iterative method). We obtained for both layers (with BB origin) the following IP values: 40 and 72 sr for LR@532,
20  2.3 and 0.9 for EAE, 1.7 and 1.8 for BAE@532/1064. As seen, the value of BAE@355/532 and LR@355 could not be estimated due to unreliable profile for backscatter at 355nm. However, we suspect that the backscatter profile at 532 nm is not accurate either (we do not know how ABR=1.2 was chosen).

**Fig. 5e).** Both layers were identified as having BB origin. The following IPs values were calculated: 20 and 31 sr for LR@355, 1.1 and 1.7 for EAE.

25  **Fig. 5f).** Only the second layer was identified as having BB origin. The following IPs were determined. LR@355=46sr, LR@532=91sr, EAE=0.25, BAE@355/532=1.9, BAE@532/1064=0.9, PDR@355=2.8%, PDR@532=2.2 %.

**Fig. 5g).** Only the first layer was identified as having BB origin. The following IPs were calculated: BAE@355/532=1.9, BAE@532/1064=1.3. Due to various criteria, PDR was not estimated.

**Fig. 5h).** This dataset was eliminated as considered to have no BB origin.

30  **Fig. 5i).** All layers have BB origin. The following IPs were calculated: 21sr for LR@355 (Ist layer), 67sr for LR@532 (Ist layer), 1.2 and -0.56 for EAE (Ist and IInd layer).

Note that all the profiles showing regions with PDR355 > PDR532 belong to the same event (long range transport from North America), recorded over 7-10 July 2013 in Warsaw, Belsk and Cabauw (discussed in part II). PDR is provided only by Warsaw ("waw") for this event and it is in accordance with results reported by Janicka et al. 2017. The PDR values retrieved for the
35  entire three days period are small and very close in value, slightly larger for PDR355. Thus, for eight cases where we had estimates at both wavelengths, we obtained the mean and STD as: PDR355 = 2.5 ± 0.5 % and PDR532 = 2.4 ± 0.9 %.

We would like to mention again that it is hardly possible to thoroughly check each profile manually in full detail because of the very high number of profiles analysed (> 4000 profiles) and also to check how the individual retrievals were performed. We only have access at the final product (optical properties). It was behind the scope of this study to check how the raw data
40  were processed with an in-house algorithm or SCC (to do that, firstly, raw data would have to be available; secondly, this kind of work is a study by itself and it needs huge amounts of resources). Conversely, we focused on post processing quality checks along various steps in the procedure. From these examples, we can see that many IPs were not fulfilling our QC criteria and thus rejected form analyses. We did not investigate in detail such situations when backscatter or extinction at 532 nm is larger than those at 355 nm or when PDR@355 > PDR@532. These situations are very rare but they can be real (e.g. Burton et al.,
45  2015; Haarig et al., 2018; Hu et al., 2018; Stachlewska et al., 2018). Haarig et al. (2018) and Hu et al. (2018) report PDR355

> PDR532 for stratosphere while both PDR have larger values (~20%). For troposphere, Haarig et al. report PDR355 = 2 % ± 4 % and PDR532 = 3 % ± 2 %. Haarig et al. hypothesize that the missing coarse mode in the size distribution is responsible for the high spectral dependence of PDR. Stachlewska et al. (2018) record in a layer at 2.2 - 2.4 km PDR355 = 1.6 ± 0.2 and PDR532 = 0.3 ± 0.1, this being related to advection of smoke particles.

A tremendous work was put on QC of the data analysed in this study. We do not claim it is perfect but we believe that we have eliminate most of unreliable profiles. The purpose of the Table 3 was to show how different datasets were eliminated during various stages, following various criteria.

We add the following statement at the end of section 4.1, describing the Fig. S1.

[revised manuscript text omitted]

**Anonymous Referee #3**

We would like to thank the reviewer for the thorough revision of the manuscript and for suggestions! We hope we addressed the comments accordingly.

We use across the text (below) the following highlights:

In red, reviewer's comments.

In black, our comments.

In green, the citations from the manuscript submitted.

In blue, the changed text for the revised manuscript.

In magenta, some citations from different papers.

Before answering to the reviewer's questions/comments etc, we would like to indicate a few changes that we have done to the initial manuscript. Those changes are not related with the data processing and analysing or its scientific content but rather with text cosmetics.

- added Faculty of Physics for co-author 2)

- in Acknowledgements we added Dominika Szczepanik

- in Funding, we added:

*The research was partially funded by the European Regional Development Fund through the Competitiveness 613 Operational Programme 2014-2020, POC-A.1-A.1.1.1- F- 2015, project Research Centre for Environment and Earth 614 Observation CEO-Terra, SMIS code 108109, contract No. 152/2016.*

- in References we added:

Adam, M., Nicolae, D., Belegante, L., Stachlewska, I. S., Janicka, L., Szczepanik, D., Mylonaki, M., Papanikolaou, C. A., Siomos, N., Voudouri, K. A., Alados-Arboledas, L., Bravo-Aranda, J. A., Apituley, A., Papagiannopoulos, N., Mona, L., Mattis, I., Chaikovsky, A., Sicard, M., Muñoz-Porcar, C., Pietruczuk, A., Bortoli, D., Baars, H., Grigorov, I., and Peshev, Z.: Biomass burning events measured by lidars in EARLINET. Part II. Results and discussions, Atmos. Chem. Phys. Discuss., https://doi.org/10.5194/acp-2020-647, in review, 2020.

Janicka, L. and Stachlewska, I. S.: Properties of biomass burning aerosol mixtures derived at fine temporal and spatial scales from Raman lidar measurements: Part I optical properties, Atmos. Chem. Phys. Discuss., https://doi.org/10.5194/acp-2019-207, 2019.

Müller, D., Ansmann, A., Mattis, I., Tesche, M., Wandinger, U., Althausen, D., and Pisani, G.: Aerosol-type-dependent lidar ratios observed with Raman lidar, J. Geophys. Res., 112, D16202, doi:10.1029/2006JD008292, 2007.

- Based on suggestions from Earlinet community, we changed the stations acronyms from two letters to three letters, according to the new nomenclature. This was already implemented in Part II. Thus, all the figures with references at station acronyms (on labels or title) were changed for clarity and consistency. Similar holds for tables.

- based on editor's suggestion for Part II, we changed N to North (for N America and N Africa)

- based on suggestions from the editor of Part II, we refer to the stations in the text by the full name instead of using acronyms.

- EAE355/532 was changed to EAE in the text (two instances: pp 11, line 18 and line34).

- a list with the acronyms used in the text was added in Supplement (as in Part II) for consistency. The reference at the acronyms list is mentioned at the end of Introduction.

A list of acronyms used in the current work is given in the Supplement (Table S1).

- figs. 10, 11, 12, 13 and 14 (now 7, 8, 9, 10 and 11): we added a), b) etc for each plot for an easier reference. We made small changes to the text and changed the3 figures caption:

Fig. 10 upper right plot -> Fig. 7 a

In Fig. 10, the first two left plots we show -> in Fig. 7 a-b we show

Fig. 10 lower left plots -> Fig. 7 c-d

Figure 7 caption:

Fig. 10. Measurements with the same source at Thessaloniki ('th') and Bucharest ('bu'). Event: 20140909-20140910. Left plots: (first) fires location seen during back-trajectories from each station (colour coded), (second) histogram of the fires occurrence in each geographical grid, (third) longitude and latitude of the fires' location versus fires' occurrence time, (forth) longitude and latitude of the fires' location versus measurement time at the two locations. Right plots: layers' altitude and intensive parameters for each station. Layers measuring the common fire are shown by arrows. The geographical location of the common fire is shown on histogram by an arrow.

Figure 7. Measurements with the same source in Thessaloniki ("the") and Bucharest ("ino") during 20140909-20140910. a) Fires location seen during back-trajectories from each station (colour coded); b) Histogram of the fires occurrence in each geographical grid; c) Longitude and latitude of the fires' location versus fires' occurrence time; d) Longitude and latitude of the fires' location versus measurement time at the two locations; e–i) layers altitude and intensive parameters for each station. Layers measuring the common fire are shown by arrows. The geographical location of the common fire is shown on histogram by an arrow.

Figure 11, left -> Figure 8 a)

Fig. 11, right -> Fig. 8 b)

See lower plot of Fig. 10 -> See Fig. 8 b)

Figure 8 caption:

Fig. 11. Backtrajectories and location of fires along backtrajectories within 100 km and +/- 1h for Bucharest ('bu') on 20140909 (left) and Thessaloniki ('th') on 20140910 (right). Lower plots show the altitude (a.s.l.) of the backtrajectories function of time. The fires' time is shown as well (see arrow location). The blue square denotes the geographical location of the common fire. See text for more details.

Figure 8. Backtrajectories and location of fires along backtrajectories within 100 km and +/- 1h for Bucharest ("ino") on 20140909 (a) and Thessaloniki ("the") on 20140910 (b). Lower plots show the altitude (a.s.l.) of the backtrajectories versus time. The common fire location is marked with an arrow. See text for more details.

Fig. 12, upper right plot -> Fig. 9 e

blue arrows in Fig. 12 -> blue arrows in Fig. 9 (a-e)

green arrows in Fig. 12 -> green arrows in Fig. 9 (a-e)

magenta arrows in Fig. 12 -> magenta arrows in Fig. 9 (a-e)

Figure 9 caption:

Fig. 12. LRT as measured at "at" on 20170713. First two BB layers are considered "mixed" while the third "pure NA". The arrows show the location of the fires (left plot) and the location of the smoke layers (right plot).

Figure 9. LRT as measured over Athens ("atz") on 20170713. a) Location of the fires; b) Histogram of the fires. The North America fires are marked by arrows; c) Fires' coordinates versus fires' occurrence time; d) Fires' coordinates versus measurements time; e) Location of the layers marked by arrows. First two BB layers are considered "mixed" while the third (magenta) "pure NA"; f) – i) Intensive parameters.

The trajectory layer in Fig. 13 is the higher one (light blue). -> The trajectory layer in Fig. 10 c) is the higher one (light blue).

Figure 10 caption:

Fig. 13. Backtrajectories for layers shown in Fig. 12 for Athens ('at') station. First two layers are considered "mixed" while the third is considered "pure N America". See text for more explanation. The squares show the location of the fire.

Figure 10. Backtrajectories for layers shown in Fig. 9 for Athens ("atz") station. Layers in a) and b) are considered "mixed" while the layer in c) is considered "pure N America". See text for more explanation. The fire location is marked with an arrow.

Figure 11…. Upper plot shows the location of the fires…-> Figure 11… a) Location of the fires

Figure 11… The bottom plot shows the histogram of the fires -> Figure 11… b) Histogram of the fires

Figure 11 caption:

Fig. 14. SE Europe region formed by stations "at", "bu", "po", "sf" and "th". Upper plot shows the location of the fires detected by each station. Note that due to overlap some are not seen. The bottom plot shows the histogram of the fires detected by each station.

Figure 11. SE Europe region formed by stations Athens ("atz"), Bucharest ("ino"), Potenza ("pot"), Sofia ("sof") and Thessaloniki ("the"). a) Location of the fires detected by each station. Note that due to overlap some are not seen. B) Histogram of the fires detected by each station.

Figure 12 caption:

Fig. 15 Scatter plots between various two intensive parameters for SE region. The colour code of the points is station related (as labelled in the title). The colour code for the mean and STD values is related with the source origin (as stated on the plots).

Figure 12. Scatter plots between various two intensive parameters for SE region (LR@532 vs LR@355, LR@532 vs PDR@532, EAE355/532 vs BAE355/532, EAE355/532 vs BAE532/1064, EAE355/532 vs LR@532 and BAE532/1064 vs BAE355/532). The colour code of the points is station related (as labelled in the title). The colour code for the mean and STD values is related with the source origin (as stated on the plots).

A list of acronyms used in the current work is given in the Supplement (Table S1).

Pp 4, line 18, pp 6, line 22: change 60 % to 65 %.

- Based on editor suggestion for part II, we do not discuss any more in Part II the other event for "common fire" analysis (here section 5.1). Thus, we will add few comments here at the end of section 5.1. Note that it was an error on the manuscript: we mistakenly wrote 20150602 instead of 20170602.

Initial: „For the other event with common IP for the same source (20170529-20150602), the smoke was labelled as of 'single fire' as no other fires were identified along the backtrajectory. This event will be discussed in the subsequent paper."

Changed: For the other event with common IP, recorded in Athens and Thessaloniki during 20170529-20170602, the smoke was labelled as of 'single fire' as no other fires were identified along the backtrajectory. The common fire occurred on 26[th] of May at midnight in Ukraine (48.171 N, 30.622 E) and it was recorded in Thessaloniki and Athens on 29/05 and 31/05 respectively. BAE@532/1064 value in Thessaloniki was less than half of that in Athens, while BAE@355/532 was larger for Thessaloniki. High BAE corresponds to higher backscatter at smaller wavelengths, which indicates a higher number of small particles. The values in Thessaloniki correspond to a higher number of small size particles (at 355 nm) and with a higher proportion of large particles (at 1064 nm) compared to the ones over Athens. $CR_{BAE}$ (colour ratio of the backscatter Ångström exponents) increases from Thessaloniki to Athens (0.22 to 0.78, respectively), which suggests an increase with travel distance (time). As $CR_{LR}$ (colour ratio of the lidar ratios) and EAE (extinction Ångström exponent) were not available to characterize the smoke in terms of age, we classified the smoke as aged based on the duration of the travel time.

Overall, we conclude that the number of common events as well as the number of the common IPs is limited and, thus, no thorough examination of these events is possible. The most important feature of this analysis is that it enables us to quantify

the smoke as of 'single fire' or 'mixed' and hence explain various IP values. This kind of analysis can be successfully applied in the future, when more data become available.

**Answers to specific comments of the Referee:**

5 … However, in the current form I still see room for improvements, both from technical and conceptual aspects.

Let me start from the principal point of view. For a publication in ACP the described method should be so clear, novel and comprehensive that the same methodology could be applied elsewhere and stands out as an independent result. Reading the manuscript in the current form I was wondering, whether a publication in a data journal with a sound description of according metadata would be an alternative.

10 We committed to bring a contribution to the special issue for EARLINET. As the special issue in AMT was closed, we agreed to submit the manuscripts in ACP (where similar topics, based on algorithms or methodologies were already published, e.g. in this issue: Jimenez et al., 2020; Wang et al 2019, **Papagiannopoulos et al., 2018;** Nicolae et al., 2018 - https://acp.copernicus.org/articles/special_issue834.html). Currently, we are constrained with time and manpower and therefore not able to entirely rewrite, adjust and resubmit the manuscript to another journal. Meanwhile, the Part II paper was
15 also submitted to ACP, the same special issue.

In any case the authors may consider to publish the data in itself. My feeling is that this will be worth it (see below).

We will indeed consider the publication of the selected slot of data and the obtained data products itself in the future. For this we plan to approach Lucia Mona, who has experience in publishing specific Earlinet data slots in repositories.

From a technical point of view the error estimation should be improved. In the manuscript (see my detailed remarks below)
20 the authors state to have a SNR of 2 as a limit and derive astonishingly small error intervals for intensive quantities. To me this does not fit together. I am convinced that a trustful evaluation of backscatter with SNR >=2 is on the edge, but the derivation of extinction or intensive quantities will be impossible. Maybe the SNR > 2 refers to raw data of individual shots, but the final analysis is done with a coarser resolution? Please make this clear.

**The input data in analysis are aerosol backscatter, extinction coefficients and particle linear depolarization ratio (along**
25 **with associated errors). This data is provided into the Earlinet database in a so-called b-files and e-files. Thus, we do not derive any of these variables or their errors.** First, we used SNR < 2 to reject the mean values of the optical properties in the layers (backscatter, extinction, PDR). Second, we used SNR < 2 to reject IOPs (LR, EAE, BAE). We used error propagation to compute the mean optical properties in the layers and then IOPs. Only during statistics, we computed STD (as stated in the manuscripts). The data in b-files and e-files represents one profile which is the average over dozens of minutes
30 (usually 1 h) and then followed by some smoothing (corresponding to "resolution evaluated"). In most of the cases, the spatial resolution for "resolution evaluated" is identical with the raw resolution (e.g. 7.5 m). During this process, the errors decreased. As mentioned, we do not have the whole picture of how the average profile was computed and smoothed nor how the errors were computed.

One possible usage of this paper might be by researchers outside the lidar community for further analysis and then a trustful,
35 transparent error analysis will be mandatory.

The error analysis in this study is very simple. Thus, the input errors associated with each optical property profile (provided in b-files and e-files), are further propagated using error propagation. For the scatter plots and other statistics studies performed in part II paper we used STD (clearly stated).

Further, there is hardly any description of underlying lidar data from the EARLINET data base. I was wondering, looking at
40 BAE in Fig 15, whether all lidar stations are using the same boundary condition to assess the backscatter?

As a principle, the boundary calibration for all stations in done with the classical approach – calibration in the aerosol-free range with guessed backscatter value. Therefore, it is not possible that precisely the same boundary condition is used at each station. The boundary condition for backscatter is retrieved by individual stations (with their own algorithms) or, when the retrievals are performed by SCC (Single Calculus Chain), there are several criteria involved. For SCC, we cited D'Amico et

al, 2015; D'Amico et al, 2016; Mattis et al, 2016. See also Pappalardo et al. (2014) who describe Earlinet (see. 3.4 Database content). In Mattis et al. (2016) it is mentioned:

"According to the EARLINET requirements, errors of backscatter coefficients at 355 and 532 nm (1064 nm) have to be below 20% (30 %) or smaller than $5x10^{-7}m^{-1}sr^{-1}$ (Matthias et al., 2004)."

5  Users have several options to choose when retrieving the optical parameters. For example, regarding the region of retrieval (Mattis et al, 2016):

"Both, minimum and maximum altitudes are to be provided by the user for each individual product."

ELDA (EARLINET Lidar Data Analyzer) searches automatically for a calibration region where backscatter coefficient has a minimum (or backscatter ratio is 1).

10  "A calibration window of user-defined width is shifted through the altitude region, where particle-free conditions typically occur (user-defined calibration interval)."

Regarding Fig. 15, here we have plotted BAE. We specified the filtering process for IPs (pp 11, lines 12-13) where we dismiss the BAE data below -1 and above 3.

To me, one of the overarching questions (not necessarily to be answered by this paper alone) is "to what extent the information
15  content of lidar data is sufficient to track changes in the microphysical properties of (BB) aerosol". I assume that the existing data set, presented in this work, is the best and most complete from quality-assured ground based stations we have. Further immediate applications would be to not only apply a fire mask but generally an aerosol emission map to further distinction between pure BB from mixed events. Also, soil and vegetation maps may be included (In the future).

Thank you for this comment! Indeed, it would be very interesting to assess if we can track changes in the microphysical
20  properties of (BB) aerosol. For such study, it is important to use the data of the stations that are able to provide input data for the inversion algorithm in each BB layer. We did not attempted a study of the microphysical properties of (BB) aerosol because in spite of the large dataset from 13 stations over 10 years ('best and more complete dataset over 2008-2017'), the number of events where simultaneously 3 backscatter + 2 extinction + 1(2) depolarization data (regarded as optimal input for microphysical inversion) is rather strongly limited. Note that Janicka & Stachlewska 2019 (ACPD, this special issue), derived
25  the optimal data set for such study, where the optical properties were derived on a fine scale specifically to address microphysical inversion. However, in our study, being a consequence of the strict QC and the rigorous fire search, the final IPs dataset with 3+2+1(1) was too small (a few cases). Thus, at this stage was not possible to perform a thorough analysis over microphysical properties. However, a more complete data set should be available now (with the additional contribution from the last 2-3 years). At this stage it is not feasible to answer about the changes in microphysical properties. However, it is indeed
30  worth trying in the future. Note that we aimed to analyse the changes in optical properties by considering several stations measuring the same fire smoke but still had difficulty to gather enough IPs for our BB cases. Anyhow, the approach of "common fire" can be successfully applied for the future (more complete) datasets and both optical and microphysical properties can be studied.

Finally, without disclosing much details, we confirm that we continue the research and already started looking further into
35  vegetation type (where the fires occurred). Preliminary results will be shown during ELC 2020. However, so far, we did not consider emission map or soil map, this may be done in future though.

A statement that a fire occurred in country XY at coordinates zz can only be the starting point for a deeper analysis. At the end, the question will be what type of vegetation on what kind of ground is burning at a shortage or not of oxygen in dry or moist conditions. These factors will probably later on determine the values of the intensive aerosol parameters. Such a work
40  can clearly not be done by the authors here but shows the value of their data set.

As said, we staring looking into vegetation type. We did not consider yet the soil type or emission map. Thank you for suggestions!

In section 5 the underlying hypothesis could be sharpened, e.g. by stating that with time / travel distance / max. encountered humidity the AEA, BAE, PDR may change like that : : : and for this reason the North American events may stay out like : : : or in the scatter plots the quantity xxx is plotted vs. yyy

We did not consider humidity in this analysis. Such work was done by Janicka & Stachlewska 2019 (ACPD, this special issue). However, for a specific case in part II paper, we considered indeed the RH values for the results interpretation.

For the scatter plots in section 5, we did not mention X vs Y individually as we referred to all combinations. On page 17, line 8, we have:

[revised manuscript text omitted]

So my opinion on this paper is mixed: provided that the error calculation is revised and honestly be stressed what can or cannot done with this data set, the scientific significance of the data set is very high. In its current form the papers is lacking substantial novel conclusions, though. Due to the principal importance of this data set and the work the authors performed so far to create

it I rate the scientific significance as "excellent" (provided the error treatment will be revised). Due to the overall quality of the paper in its current form I rate the scientific quality as "low".

In the Summary and conclusions, (pp 18, line 13) we stated:

…, the quality checks remove a large amount of data (more than half of the initial dataset).

Regarding the dataset "potential", we mention the following. Table 1 shows the data files input along with the corresponding optical properties. As said, there are 2341 files QC by Earlinet (from 3579, i.e. 65%) and 1248 new or reprocessed files (not QC by Earlinet). When working with Earlinet dataset, one has to consider the files mentioned in QC_2.0 (which passed both QC criteria) which here, represented ~ 65% of the total number of files available. See section 3. The 3589 files (2341 + 1248) analysed correspond to 1138 time stamps. **Table 3 shows how this dataset (1138 time stamps) squeezes along various criteria involved in QC or other checks** (such as fire origin of the pollution layer). Columns 3 and 4 show the dataset after in-house QC of the data and the retrieval of the layers (as layers could not be retrieved for all profiles). Columns 5 and 6 show the chosen data as having BB origin (fire along backtrajectory). Columns 7 and 8 shows the restrained dataset after imposing SNR > 2 for optical properties. Finally, columns 9 and 10 show the final dataset used for data interpretation in terms of IPs, after imposing SNR > 2 for IPs along with a filtering of the IPs data. These steps (with numbers involved) were commented in the text. Thus, if we do a relative difference between columns 1 and 2, we observe that overall, 15.6 % of the initial number of time stamps was rejected due to in-house QC or absence of pollution layers. Further, 29.5 % of the time stamps were eliminated as being considered of non-BB origin (according to our criteria). This corresponds to 44.6 % layers rejected. SNR > 2 for optical properties eliminated just one time stamp (0.1%) and 3 layers (0.3 %). The last criteria rejected 22.2 % of the time stamps and 24.3 % layers. These differences are with respect to previous step. Thus, overall, we have rejected 45.2 % of the time stamps (column 9 with respect to column 3) and 58.2 % of layers (column 10 with respect to column 4). For time stamps, this is on top of 15.6 % time stamps eliminated during QC and estimation of pollution layers. Thus, we have 53.8 % of the time stamps eliminated. As described in section 2.1, Earlinet QC rejected ~ 35 % of the files submitted. Note that we mistakenly wrote ~ 60 % instead of 65 % for selected files. We corrected the value. Our numbers in table 3 refer to the number of time stamps (not to the number of files) and thus it is not straightforward to estimate the number of files rejected. As seen in Table 1, for 1138 time stamps there were 3589 profiles [as for one time stamp we can have from 1 optical property profile (Sofia) to 7 optical properties profiles (Warsaw)]. We did not check how many time stamps corresponded to the rejected files (we did not look at all at those files). For a simple calculation, if we extrapolate from 1138 time stamps/3589 files to x time stamps/1238 files rejected we obtain ~ 393 time stamps eliminated by Earlinet QC (~ 26 %). Thus, overall, we would have 526 time stamps/1531 total time stamps, i.e. a rejection of ~ 66 % for the time stamps. It is not so easy to extrapolate to the total number of files rejected. If we extrapolate 35 % rejected data from 3759 Earlinet QC to the total number 4827 files (3759 Earlinet QC + 1248 not-Earlinet QC) then we obtain a rejection of ~ 47 %.

Figure S4 (former 8) shows an example with the IPs distribution for all the events (the graphs for all stations are shown in Part II). One can see in the plot that there are quite a limited number of events where we have all IPs determined (not considering PDR though).

More discussions are present in Part II, where we also mention what can we do and not. For the examples in Part I, one can see the following:

-for Fires events seen by two stations (5.1): the number of IPs to be compared is limited. Thus, for the example chosen here, only BAE@532/1064 can be compared. For the other example with common IPs, the two BAE could be compared. As said, the most important IPs to characterize the aerosol are EAE, $CR_{LR}$ and PDR.

-for LRT case study (5.2.1), again, the two BAE were the common IPs for the 3 layers analysed. The second layer had the two LR available while the third layer had also EAE estimated.

- the scatter plots between various IPs specific for a geographical region (5.3.2, here SE Europe) is aimed to show a potential pattern between them, based on different continental source origin. As observed, not always we have data for all continental sources which we aimed to find a specific pattern. Part II shows the scatter plots for the other regions. Then we characterize and compare each region function of source origin.

Thus, these examples, for each type of investigation, show that in order to better characterize the smoke, more IPs are needed (ideally, all). In many cases, this was not possible. We would like to emphasize though that the methodology, including the types of investigation is quite robust and can be successfully applied when data proving more IPs is available.

5   A few novel approaches were considered in this study: the smoke layer should have at least one fire along backtrajectory, the quantification of the number of fires contributing to a smoke measurement [here we classified the smoke as having the origin from one fire or more fires (mixed smoke)]. We also classified the smoke from North America as originating solely from North America or being mixed with local fires. Lastly, based on statistical analysis, we divided the stations in four regional clusters and we classified the smoke as having the origin in different continents or many continents (mixed smoke). We regret that these were assessed by the Referee as only "low" scientific quality. However, we agree that the results were not as spectacular
10   as we initially expected (the main reason being the small final dataset for IPs).

**Some specific remarks are:**

Intro line 19: what do you mean be fuel with respect to climate change?

The report states: "Climate change affects forest fires both directly through the weather conditions that affect fire ignition and propagation, and indirectly through its effects on vegetation and fuels." Fuel refers to the biomass which can burn (combustible
15   material). See types of fuel as type of the biomass on fig. 107 of the study: agricultural field crop, broadleaved forest, grassland, needle forest or mixed. We agree the citation is not clearly stated. We rephrase as:

Initial: As mentioned in the report, climate change affects forest fires through the weather conditions, vegetation and fuel.

Changed: As mentioned in the report, climate change affects forest fires through the weather conditions and through its effects on vegetation and fuels (combustible material),

20   Fig 2, No.8 what are the limits for neglecting intensive parameters, what percentage of data has been rejected?

First, we apologize we forgot to mention "Fig. 2, stage VIII in Methodology" when talking about the calculation of the intensive parameters. Thus, on page 11 we introduce:

Initial: Once the mean optical properties were calculated, the intensive parameters were determined (where possible).

Changed: Once the mean optical properties were calculated, the intensive parameters were determined, where possible (Fig.
25   2, stage VIII in Methodology).

We mentioned in the text the condition SNR>2 and then the limits (pp 11, lines 12-13):

All the IPs have SNR > 2. The imposed limits (data filtering) for IPs are the following: LR@355 = [20 150] sr, LR@532 = [20 150] sr, EAE = [-1 3], BAE@355/532 = [-1 3], BAE@532/1064 = [-1 3], PDR@355 = [0 0.3] and PDR@532 = [0 0.3] (following closely Burton et al., 2012; Nicolae et al., 2018).

30   The percentage of the rejected data can be calculated from Table 3 by comparing columns 9 (# time stamps) and 10 (total # layers) with columns 7 and 8 (either by station or per total). On the same page 11, lines 31-34 we mention the rejected number of values for each IP.

The number of outliers dismissed based on predefined ranges of acceptable values for each intensive parameter is small (3.7% per total). For each IP we have the following numbers: 8/305 (2.6%) for LR@355nm, 8/253 (3.2%) for LR@532nm, 18/243
35   (7.4%) for EAE355/532, 39/642 (6.1%) for BAE355/532, 21/706 (3%) for BAE532/1064, 0/132 (0%) for PDR@355 and 0/242 (0%) for PDR@532.

Section 4 P 7, l 13: with kappa you mean the extinction, right? Before you have used the symbol "e"

Yes, we use kappa for extinction. We add the list of acronyms in Supplement as we did for Part II. Thus, at the end of Introduction we add (similar with Part II):

40   A list of acronyms used in the current work is given in the Supplement (Table S1).

Symbols „e" and „b" are used only for the data files, as they are stored in Earlinet database (no Greek symbols were used in file names).

L 20: is there a reason for the bin numbers? (the product of resolution * numbers is not constant) how does this different resolution later affects you layer selection?

The choice of the bin number is rather based on trial and error, looking at the input profiles. Please note that even if the raw resolution mentioned in the file was 3.75, 7.5, 15, 30 or 60 m, the smoothed resolution ("resolution evaluated") for the retrieved data (extinction, backscatter and depolarization) was very different among stations (maybe even different for the same station). As mentioned in the text it was almost impossible to extract the information about how the errors were calculated and how the smoothing was performed. Some information is given as a (free style) comment in the general attributes but it is almost impossible to quantify it as we dealt with few thousands of profiles which can have hundreds of free style comments. Please note that this smoothing in this stage is used only to determine the layers. The optical properties are calculated based on input data as it is in the netCDF file. As mentioned, the automatic retrieval for layers boundaries has limitations and that's why we double check visually the layers. Where there were non-accurate boundaries, we adjusted them "by hand".

P8, L 4: I do not understand the possibility that a maximum is not surrounded by 2 minima. findpeaks is employed for beta1064, i.e. real data, so I assume intervals with constant values are at least not frequent?

Sometimes there is the possibility that a maximum is not surrounded by two minima. Theoretically, there are two minima but the criteria employed with the function *findpeaks* can eliminate one of the minima (e.g. criterium at line 3). In example 5i) we can see that in layers 2 and 3 there are more local maxima and minima but finally only one maximum is chosen.

I am not sure what you refer to when mentioning intervals with constant values. There are intervals with more or less constant values. As it is almost impossible to have identical values over an interval, the function can see more (local) maxima and minima. However, adjacent maxima are eliminated (as said on line 1) based on "prominence width'. Further, the algorithm can determine the minima behind the two maxima. In the worst case, we adjust manually.

P8, L 7: in which way this last criterion is independent from the others (and hence needed)?

You are right, it looks redundant. We eliminate this sentence and we add in parenthesis the following to the previous statement:

Initial:

- a maximum peak should be bordered by two minima; when the first or the last minimum is missing, a criterion is used to add the missing minimum; thus, the minimum is chosen at a location (> 300 m from the first or the last maximum peak) where the optical property has the minimum value

Changed:

- a maximum peak should be bordered by two minima (which defines the layer boundaries); when the first or the last minimum is missing, a criterion is used to add the missing minimum; thus, the minimum is chosen at a location (> 300 m from the first or the last maximum peak) where the optical property has the minimum value

L24/25: "Our approach : : : in line with : : :" this statement is completely unproven. Maybe you do not need it but to prove it you should apply your code to one of the examples of the mentioned papers.

The comparison with different authors was based on visual inspection of their retrievals for layer's boundaries. We did not intend to apply the algorithm to those examples (not even think about!). It is not a matter of an algorithm but rather a matter of personal choice/criteria on how to choose the boundaries (which is the input for the development of an algorithm). Recall that in general, the research papers deal with case studies and in most of the cases, the boundaries are chosen by visual inspection.

P9 L 15 and 16: where is table S2? In my version of the manuscript the citations are not numbered.

Table S2 (now S4) is in Supplement (now page 12).

P10, L 24 I do not understand the definition of data availability: Isn't the ratio between layer thickness and resolution simply the number of data points? Did you use an additional criterion to flag a given data point as valid?

The number of data points in the layer is indeed the ratio between layer thickness and resolution. In order to evaluate (calculate) the mean, we impose (criterion) that 90% of the points to be available. The confusion might come from the fact that we have 100% of the points are available for the profile used to determine the layer's boundaries (e.g. b1064) but we might not have 100% available for the other profiles (other backscatter and extinction). In general, the extinction profiles are shorter. In order to compute an intensive parameter in a layer, we should have ~ the same number of points available in both backscatter and extinction otherwise we do not calculate over the same structure (layer).

The points which are not valid in a layer are simply NaN in the data file. Thus, NaN values should be below 10%.

P11, L 11 and following: is SNR >=2 really sufficient to determine particle extinction and lidar ratio? The SNR of the underlying Raman channel must have been much better, I assume. Fig 9 is interesting. However, presented like this: almost spherical particles, very high LR and very small particles I am wondering whether this is really BB or anthropogenic pollution. It would be interesting to show the hysplit trajectories for the extreme cases (maybe 2015, 2016, 2017).

**The particles extinction, backscatter and depolarization are provided in the data files with associated errors**. The Earlinet database is not provided with the details involved to calculate and validate the optical properties (including errors). By definition, they are Earlinet quality checked and approved by PIs once they are in the database. Additional quality checks can be performed by users.

As seen in Fig. 1 (methodology scheme), one can see that the SNR is not referring the lidar signals (which indeed with SNR of 2 likely would not be sufficient to obtain optical properties). As in Fig.1, we apply this criterion on profiles of optical properties not signals, whereby at first, the mean values of the optical properties derived in the layers containing BB are kept only if SNR>2. Then, we calculate the intensive parameters and keep them for analyses if their SNR>2.

We have difficulty to understand what you refer to when talking about Fig. 9. This figure shows all the data available (in terms of IPs) for this station. As we can see, the min-max values range for each IP (as reported in literature) is quite large and surely it depends on the measurement site and characteristics of particular BB event and thus we have both small and large values for each IP.

As for the IPs in Warsaw, they are quite well spread within the limits, except for PDR. However, taking into account that upper limits reported in literature of 0.2 at 532 and 0.38 at 355 are very high and more in the range of values expected for mineral dust particles not smoke particles. For BB events reported at Warsaw station the PDR in general has a lower upper values limit that this reported in literature (i.e. below 0.18 at 532 and 0.12 at 355), hence here BB shape is closer to spherical particles. The LR and EAE spread over a large range of values, and thus we can have both smaller (high EAE) and larger (small EAE) particles. Unfortunately, we did not investigate the potential contribution of other types of pollution in these layers (it would have been another enormous work to be done for 1901 layers) and we assumed as being of BB origin only. Also, for the values reported in literature – we had no means to investigate to which extend the BB events were contaminated e.g. with mineral dust or pollution.

Note that the lower limit for literature values for BAE@532/1064 was missing in Fig. 9. **We added it.**

For a better explanation (hopefully) we changed the following:

Several values outside the literature range are observed for EAE, both BAE and PDR@355 for Warsaw.

In general, LR, EAE, both BAE and PDR are spread well within the reported limits, although the latter is rather in its lower range. Lower PDR may be explained as due to specific location of Warsaw being much closer to BB sources in Ukraine, Belarus and Russia than other EU countries, and therefore much more exposed to faster and more direct BB transport to this site. In such case, differences of properties with respect to other measurement sites in Europe are expected and revealed within our work. Moreover, Warsaw is much less exposed to mineral dust intrusions, in comparison with many EU sites at which the BB measurements are more likely to be affected by slight contamination of dust, e.g. Spanish sites. On the other hand, Warsaw is an urban site, thus for BB observed in layers at lower ranges a slight contamination of the BB with the local urban pollution is possible. For higher layers, industrial pollution form Silesia region in Poland (Stachlewska et al. 2018) or Ruhr region in

Germany (e.g. case here on 20160704 at 07:30) is possible. This can explain a few values outside the literature range observed in Warsaw, i.e. for EAE (2 values), for both BAE (33 values) and for PDR355 (8 values). The extreme values for EAE observed on 20160704 07:30 and 20170619 20:30 (2.6 and 2.8 respectively along with $CR_{LR} < 1$) correspond to fires in Germany / Belgium and United Kingdom respectively. The fires occurred in less than 40 h before smoke measurements and thus we can consider the smoke relatively fresh.

We are not sure to which extreme cases you refer to and thus we try to guess. Looking at the two EAE extreme values in 2016 and 2017, we checked the backtrajectories.

The two extreme points for EAE from 2016 and 2017 corresponds to the time stamps:
1) 20160704 07:29:30
2) 20170619 20:29:30
Note that the times are the middle of the time interval (so the input files have a time with ~ 30min before).

Both events had two layers but for the second layers not all IPs were retrieved.
These are the IPs values:

| Altitude | LR355 | LR532 | EAE | BAE355/532 | BAE532/1064 | PDR355(%) | PDR532(%) |
|---|---|---|---|---|---|---|---|
| 1) | | | | | | | |
| 2088m | 114+-/6 | 65+-/3 | **2.6**+-/0.15 | 1.2+-/0.05 | 0.87+-/0.03 | N/A | 3.3+-/0.03 |
| 5095m | N/A | N/A | N/A | N/A | 1.7+-/0.05 | N/A | |
| 2) | | | | | | | |
| 2417m | 64+/-2 | 35+-/3 | **2.8**+-/0.2 | 1.32+-/0.07 | 0.84+/-0.04 | 1.65+-/0.02 | 3.2+-/0.04 |
| 3306m | N/A | N/A | N/A | 1.64+/-0.04 | 0.94+/-0.03 | 1.5+/-0.01 | 2.5+-/0.02 |

Below are the FIRMS-Hysplit plot for these events. The two extreme EAE values correspond to the first layer.

Looking into firms-hysplit plots we see:
1) The fires along backtrajectory for the first layer are located in Germany and Belgium and are detected somewhere between 24h and 48h back. The second layer 'detects' one of the fires seen by first layer but after ~192h.
2) The fires along backtrajectory for the first layer are located in UK and are detected somewhere between 24h and 48h back. Actually, both layers 'detect' the same fire at ~ the same time. The fire is closer to the second backtrajectory (second layer).

Taking into account that the fires are seen somewhere after around 36 h back we can say it is relatively fresh (and thus high EAE and $CR_{LR}<1$). The other IPs are withing literature limits.

[Figure]

[Figure]

Plot above: Illustration of Event 1) 20160704 07:29:30

[Figure]

**wa-20170619T200000-20170611T000000-Hysplit+FIRMS**

FRP<100  100<FRP<500  500<FRP<1000  1000<FRP<2000  2000<FRP

Plot above: Illustration of Event 2) 20170619 20:29:30

P13 L2: Sorry, I missed that: why PDR (at 532) increases with time? I would have assumed that coagulation makes the particle shape more spherical

The statements at the beginning of the page 13 are based on literature review, discussed on the previous page. However, we erroneously cited Veselovskii et al. (2018) where PDR@532 is shown as increasing while EAE decreases (a decrease of EAE being related with longer travel time). Revising their plot (Fig. 10) we realize that it is based on data from both smoke and dust while EAE range is between 0 and 0.9. At a more careful reading, PDR@532 from smoke is below 10%. Linked with corresponding EAE on the plot, we do not see any correlation for smoke. We deeply apologize for this misunderstanding. On the contrary, we have cited Nisantzi et al. (2014) in Part II which reported PDR@532 decreasing with time. We remove the statement and add the reference to Nisantzi et al. (2014). However, the entire paragraph is rephrased as suggested (shown above).

Remove: PDR@532 increases with time (while EAE decreases).

Changed: PDR@532 decreases with time (Nisantzi et al., 2014).

P13 L 6 (and P14 later) I would omit mentioning Fig 2 for things which the reader cannot verify on his own.

We are sorry but we did not understand the comment. In Fig. 2 we mentioned the most important steps in methodology. From stage IX we refer to specific analysis indeed. The intention was to point out to a specific stage of the methodology when we start a new section (discussion).

P14, L 7-8 the errors in the IP are unbelievable small and non-proven. If this is the error of many different measurements, you rely on the fact that the aerosol layer did not change during the sampling. Contrary, the different BAE between "bu" and "th" are much more understandable, if the fire conditions, the heat, the burning plants the amount of water vapour : : : etc changed.

Indeed, the uncertainties are quite small. The IPs have the errors propagated based on uncertainties provided for the optical properties. The optical profiles represent one single profile over 1-h measurement. According to the station PI, the air mass sampled during this one hour seems not to change. Also, it has to be noted that the uncertainty is calculated without considering the systematic biases (e.g. the uncertainty of the estimated constant lidar ratio over the profile).

In general, we have observed small uncertainties for many IPs and many stations. This fact can be seen in Part II where all the data are shown in Supplement. The mean, median, min and max values for IPs, for each station, are shown in Part II (Table 1). There, we can see that the values of the uncertainties for BAE (median and minimum) for Bucharest station are among the smallest.

To double check, we have recalculated the values for BAE for the example in Fig. 10. We obtained the same values for both stations.

We mentioned in 2.2 (metadata), pp 5, lines 24-25 that "we could not determine which was the final spatial resolution nor the method used for smoothing and error calculation".

For this example, for Bucharest station, the information provided in the attributes of the netCDF files about effective resolution is vague ('resolution evaluated': 'Linearly increasing from 7.50m to 1500.00m'). For Thessaloniki case, the data processed with SCC provide a new variable called 'VerticalResolution'. The vertical resolution for this case is 7.5 m for 355 nm and 532 nm and 15m for 1064 nm everywhere, i.e. the same as for raw data resolution. Basically, there is no smoothing here and thus, we expect larger uncertainties as compared with smoothed profiles for Bucharest. In addition, there were 35400 shots and 18000 shots averaged for Bucharest and Thessaloniki respectively.

As mentioned in the manuscript, it is very challenging to work with thousands of data files from 14 stations, using 14 different algorithms, several methods for deriving optical properties, many ways to smooth data and to choose the resolution and other many ways to compute the errors. The Earlinet community keeps improving SCC algorithm and thus, all the data will be treated in the same manner in the future. However, keep in mind that the user still has to add few input information (e.g. min and max altitude to be used, windows for searching the calibration region). Earlinet has a tool for QC, the PI has to approve the data before uploading to Earlinet. Still, any user which uses Earlinet database does his/her own QC. We did our best to reject suspicious data in this study.

**1 List of acronyms**

**Table S1. List of acronyms**

| Nomenclature | Definition |
|---|---|
| ACTRIS | Aerosol Cloud and Trace Gases Research Infrastructure |
| a.g.l. | Above ground level |
| a.s.l. | Above sea level |
| "atz", "brc", "cog", "ino", "cbw", "evo", "gra", "lei", "mas", "hpb", "pot", "sof", "the", "waw" | Athens, Barcelona, Belsk, Bucharest, Cabauw, Evora, Granada, Leipzig, Minsk, Observatory Hohenpeißenberg, Potenza, Sofia, Thessaloniki and Warsaw (lidar stations considered in this study). |
| BAE | Backscatter Ångström exponent. $BAE@355/532=-\log(\beta p355/\beta p532)/\log(355/532)$, $BAE@532/1064=-\log(\beta p532/\beta p1064)/\log(532/1064)$ |
| BB | Biomass burning |
| $\beta_p$ | Particle backscatter coefficient [1/m/sr] |
| CR(s) | Colour ratio(s). $CR_{LR}=LR@532/LR@355$, $CR_{BAE}=BAE@532/1064/BAE@355/532$, $CR_{PDR}=PDR@532/PDR@355$ |
| EAE | Extinction Ångström exponent. $EAE@355/532=-\log(\kappa p355/\kappa p532)/\log(355/532)$ |
| EARLINET | European Aerosol Research Lidar Network |
| EU, AF, NA, AS | Europe, Africa, North America, Asia continental source regions |
| EUAF, EUNA, EUAS | Europe + Africa, Europe + North America, Europe + Asia continental source regions |
| FIRMS | Fire Information for Resource Management System |
| FRP | Fire radiative power |
| GDAS | Global Data Assimilation System |
| HYSPLIT | Hybrid Single-Particle Lagrangian Integrated Trajectory model |
| IP(s) | Intensive parameter(s) |
| $\kappa_p$ | Particle extinction coefficient [1/m] |
| LR | Lidar ratio [sr]. $LR@355=\kappa p355/\beta p355$, $LR@532=\kappa p532/\beta p532$ |
| LRT | Long range transport |
| MODIS | Moderate Resolution Imaging Spectroradiometer |
| PDR | Linear particle depolarization ratio |
| QC | Quality control |
| SE, SW, CE and NE | Southeast, Southwest, Central and Northeast Europe (geographical measurement regions) |
| SNR | Signal to noise ratio |
| STD | Standard deviation |

**2 Summary of metadata**

**Table S2. Summary of main features used to calculate the backscatter and extinction coefficient, specific for each b-files and e-files. Detection mode: 1 (photon counting), 2 (analog), 3 (analog + photon counting). Evaluation mode: 1 (Klett-Fernald), 2 (Raman), 3 (aerosol backscatter ratio).**

|  | Detection mode (1/2/3) | Evaluation method (1/2/3) | Raw resolution | Shots averaged | Zenith angle |
|---|---|---|---|---|---|
| b355 | 176/153/**590** | **524**/130/257 | **most@3.75m and 60m** | **most@1e4** | **most@0°** |
| e355 | **417**/0/107 | 0/**516**/0 | **most@60m** | **most@1e4** | **most@0°** |
| b532 | 229/199/**385** | **335**/296/174 | **most@3.75m**, then 7.5m, 15m, 60m | **most@1e4** | **most@0°** |
| e532 | **409**/0/94 | 0/**496**/0 | **most@60m** | **most@1e4** | **most@0°** |
| b1064 | 222/**608**/0 | **683**/139/0 | **most@3.75m**, then 15m and 60m | **most@1e4** | **most@0°** |

**3 Calculation of the aerosol layers boundaries**

The following approach was considered to calculate the boundaries of the aerosol layers. The order of selecting the optical profile to determine the boundaries of the aerosol layers is the following: βp1064, βp532, βp355, κp532, κp355. In other words, when available, use βp1064. When βp1064 is not available, use βp532. If the latter is not available either, use βp355 and so on. Note that for the times when none of the profiles showed a pollution layer, all profiles for that specific time were excluded.

Once the optical profile (including the associated error profile) and the corresponding altitude profile are available, the algorithm developed to determine the aerosol layers boundaries is run. The steps of the algorithm are the following:

- Perform a smoothing of the optical profile. The number of bins used for smoothing depends on the input resolution. Thus, for a resolution of 3.75 m, we applied moving average over 23 bins. For a resolution of 7.5 m, 15 m, 30 m, 60 m, we used 11 bins, 9 bins, 7 bins and 3 bins, respectively. For the particular cases of "ca" and "oh" systems, we applied a number of bins of 15 and 19, respectively (as the signals were very noisy). The corresponding errors were propagated.

- Employ the function *findpeaks* from Matlab (www.mathworks.com, last access 20191126) to find the maxima, with the following options: the minimum distance between peaks is 300 m (*MinPeakDistance*) and the minimum peak height (*MinPeakHeight*) is as follows: 1e-7 for βp1064, 1.5e-7 for βp532, 3e-7 for βp355, 1e-6 for κp532 and 3e-6 for κp355. The value of the minimum distance between peaks was chosen as in Nicolae et al. (2018). If no peaks are found, the routine returns the message *no layers with maximum above MinPeakHeight*.

- Employ the function *findpeaks* to find the minima, with the following option: the minimum distance between peaks is 300 m (*MinPeakDistance*)

- eliminate adjacent maxima if the "prominence width" (https://www.mathworks.com/help/signal/ug/prominence.html, last access 20191126) overpasses the position of the adjacent maxima

- eliminate small maxima / minima peaks which are smaller than 10% of the maximum / minimum peak

- a maximum peak should be bordered by two minima (which defines the layer boundaries); when the first or the last minimum is missing, a criterion is used to add the missing minimum; thus, the minimum is chosen at a location (> 300 m from the first or the last maximum peak) where the optical property has the minimum value

Following the criteria discussed, there can be cases when it is not possible to find any aerosol layer. Consequently, those profiles were dismissed. Additionally, a manual check was performed and for the cases with non-accurate estimation of the boundaries, the boundaries were manually corrected (~ 40 % of the cases) and sometimes, we added layers which had a maximum below the threshold of the minimum peak height. Thus, we cope with a semi-automatic algorithm. Table 3 shows the number of time stamps when it was possible to determine a layer and at least one optical property could be calculated (column 3). Recall that many profiles were dismissed manually through quality check before we apply the algorithm for layer boundary evaluation and this explain most of the "missed" cases (difference between second and third columns). The initial

total number of layers, with at least one optical property (column 4) is greater than the time series (column 3) as most of the times we have more than one layer within a profile. The other columns are discussed in the next section. Overall, we were able to determine 1901 layers for 960 time stamps (out of 1138 in total).

Various authors use different criteria to estimate the layer boundaries. In most of the papers examined, the authors do not describe how they determined the boundaries of the layers. However, the boundaries can be easily identified visually (a common practice when investigating one or few cases). In a few studies it is mentioned the gradient method (Giannakaki et al., 2015; Mattis et al., 2008; Ortiz-Amescua et al., 2017; Preißler et al., 2013). When intensive parameters are available (e.g. EAE or LR), one can determine the boundaries based on intensive parameters being nearly constant in the layer (e.g. Samaras et al., 2015) or based on the ratio of elastic to Raman profiles (Vaughan et al., 2018). In situations when a few layers are visible, one can choose them as a single large layer (e.g. Ansmann et al., 2009). Our approach provides the layers boundaries in line with those shown by Ansmann et al. (2009), Janicka et al. (2017), Hu et al. (2018), Veselovskii et al. (2018).

[Figure]

a)    Two layers automatically selected based on $\beta_{1064}$ signal. Layers' boundaries are: [802 1717] m and [1717 2707] m.

[Figure]

**b)** One layer automatically selected based on $\beta_{1064}$ signal. Layer' boundaries are [1670 3426] m.

[Figure]

**c)** Three layers automatically selected based on $\beta_{1064}$ signal. Layers' boundaries are [2193 3037] m, [3037 5166] m and [5166 5809] m.

[Figure]

d)  Two layers automatically selected based on $\beta_{1064}$ signal. Layers' boundaries are [968 2002] m and [2002 3847] m.

[Figure]

5  e)  Two layers automatically selected, based on $\beta_{355}$ signal. Layers' boundaries are [1202 1862] m and [2282 2882] m.

[Figure]

**f)** Three layers selected based on $\beta_{1064}$ signal. Layers' boundaries are [915 2193] m, [2193 2962] m and [3530 4427] m. The top of the second layer was manually changed from 3530 m to 2962 m. A fourth layer around 6500m was dismissed.

[Figure]

**g)** Three layers selected based on $\beta_{1064}$ signal. Layers' boundaries are [1319 2678] m, [2678 3411] m and [9664 11846] m. The third layer was added manually. The top of the second layer was modified from 4061 m to 3411 m.

[Figure]

**h)   Three layers detected based on $\beta_{1064}$ signal with the boundaries [915 1617] m, [1617 3142] m and [5264 5801] m. The top of the second layer was manually modified from 3612 m to 3142 m.**

[Figure]

**i)   Three layers selected, based on $\beta_{532}$ signal. Layers' boundaries are [2715 3915] m, [3915 4965] m and [4965 7365] m. The bottom of the second and third layers were manually changed from 4485 m to 3915m and from 5295 m to 4965m.**

Figure S1. Examples of layers selection, based on $\beta_{1064}$ signal (a-d, f-h), $\beta_{1532}$ signal (i) and $\beta_{355}$ signal (e). Layers are shown by grey areas. All available optical properties are shown (particles backscatter coefficients $\beta_p$ on the left, particles extinction coefficients $\kappa_p$ in the middle and particles linear depolarization PDR on the right). The boundaries shown in a-e plots are the automatic output of the algorithm. In the f-i plots, one or more boundaries retrieved by the algorithm were manually adjusted.

**2. Example number of layers selected and corresponding optical parameters available for each layer**

[revised manuscript text omitted]

**4 Example number of layers selected and corresponding intensive optical parameters available for each layer**

[Figure]

Figure S4. The number of times (events) when the layers were evaluated and the corresponding number of intensive parameters available in the layer. Example for Athens station ("atz"). Layers have a biomass burning origin (fire source).

---

## Author Response (AR2)

**Biomass burning events measured by lidars in EARLINET. Part I. Data analysis methodology.**

Mariana Adam1, Doina Nicolae1, Iwona S. Stachlewska2, Alexandros Papayannis3, Dimitris Balis4

1National Institute for R&D in Optoelectronics, Magurele, 077225, Romania

[revised manuscript text omitted]
 S2+), we found the following values for CRLR and EAE for fresh, aged and NorthN America case (particular case of aged smoke). CRLR was 0.88, 1.08 and 1.23 while EAE was 1.47, 1.2, 0.95. CRBAE had the values 0.76, 0.98 and 0.98. We may speculate that CRBAE may increase with time (distance).
- 30 Please note that the time difference (an hour at most) between right plots and bottom left plot comes from the fact that Hysplit back-trajectories start at sharp hours. For example, for the measurement at 18:49 the starting point on back-trajectory is 18. Statistics over LRT from NorthN America will be shown in the companion paper. We encountered 168 measurements over the 24 periods (over 2009 2017 period). From these measurements, 77 have a NorthN America origin while the other 91 have a different BB origin (local). The LRT events from NorthN America is analysed differencing between 'pure N America'

fires (sensed smoke comes solely from North America) and 'mixed' fires (measured smoke is a mixture of North America smoke and local/European smoke).

**5.3 Analysis over geographical regions**

**5.3.1. Geographical regions**

- 5 Taking into account the position of the 14 stations (Fig. 1), four geographical regions are chosen as follows:-SE-South-East Europe (Potenza, Athens, Thessaloniki, Sofia, Bucharest"po", "at", "th", "sf", "bu"), SW-South-West Europe (Granada, Barcelona and Evora"gr", "ba" and "ev"), NE-North-East Europe (Belsk, Warsaw, Minsk"be", "wa", "mi") 
[revised manuscript text omitted]

31

20

25